# Dynamic Learning in Large Matching Markets

**Anand Kalvit[1] and Assaf Zeevi[2]**
Columbia University, New York
{[1]akalvit22,[2]assaf}@gsb.columbia.edu

## Abstract

We study a sequential matching problem faced by *large* centralized platforms where "jobs" must be matched to "workers" subject to uncertainty about worker skill proficiencies. Jobs arrive at discrete times with "job-types" observable upon arrival. To capture the "choice overload" phenomenon, we posit an unlimited supply of workers where each worker is characterized by a vector of attributes (aka "worker-types") drawn from an underlying population-level distribution. The distribution as well as mean payoffs for possible worker-job type-pairs are unobservables and the platform's goal is to sequentially match incoming jobs to workers in a way that maximizes its cumulative payoffs over the planning horizon. We establish lower bounds on the *regret* of any matching algorithm in this setting and propose a novel rate-optimal learning algorithm that adapts to aforementioned primitives *online*. Our learning guarantees highlight a distinctive characteristic of the problem: achievable performance only has a *second-order* dependence on worker-type distributions; we believe this finding may be of interest more broadly.

## 1 Introduction

**Background and motivation.** The problem of sequentially matching "jobs" to "workers" under uncertainty forms the bedrock of many modern operational settings, especially in the online gig economy, see, e.g., applications such as Amazon Mechanical Turk, TaskRabbit, Jobble, and the likes. A simpler instance of the problem dates back to [13] where it is referred to as the sequential stochastic assignment problem (SSAP). A fundamental issue in such settings is that the platform typically is oblivious (at least initially) to the skill proficiencies of individual workers for specific task categories. This complexity is further compounded by the large number of workers usually present on such platforms, tantamount to prohibitively large experimentation costs associated with acquisition of granular information at the level of an individual worker. This issue is commonly mitigated by exploiting structure in the problem (if any), or by positing distributional assumptions on the population of available workers, e.g., workers may be drawn from some distribution $\mathcal{D}$ satisfying certain context-specific desiderata. Such distributional assumptions are vital to designing efficient algorithms for these systems, and as such, traditional literature has largely relied on the availability of ex ante knowledge of $\mathcal{D}$ or certain key aspects thereof (refer to the literature review in §1.2).

**Key research question.** An important characteristic of the gig economy is that the population of workers may undergo distributional shifts over the course of the platform's planning horizon. These effects may, many a time, fail to register in a timely manner; as a result, there may be delays in tailoring appropriately the matching algorithm (calibrated typically using available distribution-level information) to the changed environment. This has the potential to cause revenue losses as well as catalyze endogenous worker attrition. Such exigencies necessitate designing algorithms that are *agnostic* to $\mathcal{D}$ and whose performance is *robust* to plausible realizations thereof.

**The model at a glance.** We consider a finite set of possible job-types (denoted by $\mathcal{J}$), an assumption we deem appropriate for settings such as those discussed above. In addition, we model workers as exhibiting discrete skill-levels (aka worker-types), indexed by $\{1, ..., K_j\}$, w.r.t. each job-type $j \in \mathcal{J}$,

36th Conference on Neural Information Processing Systems (NeurIPS 2022).

and make the simplifying assumption that $(K_j : j \in \mathcal{J})$ is known a priori. It is not unreasonable to make this assumption since it is common, in practice, for platforms to deploy pilot experiments prior to the actual matching phase in order to gather sufficient information on key primitives such as the size and stability of low-dimensional sub-population clusters, if any exist; one can therefore safely assume in settings where such structure exists that $(K_j : j \in \mathcal{J})$ is well-estimated a priori. Furthermore, assumptions pertaining to the finiteness of the set of possible job-types and segmentation of the population of workers based on discretization of skill levels are de rigueur also in the dynamic matching and the broader operations research literature, see, e.g., [7, 2, 17], etc.

While the demand is constituted by sequential job-arrivals (possibly in batches of stochastic size and composition), we posit availability of an unlimited number of workers on the supply side. The latter feature encapsulates the *choice overload* phenomenon characteristic of many large market settings where workers are available in a large number relative to the platform's planning horizon. To our best knowledge, extant literature on matching under uncertainty is largely limited to "finite" markets (see the literature review in §1.2), and therefore fails to accommodate this important practical consideration. In our setting, the population of workers, albeit large, is governed by a fixed distribution that controls the proportion of each worker-type. Specifically, the $K_j$ distinct worker-types w.r.t. job-type $j$ are distributed according to $\boldsymbol{\alpha_j} := (\alpha_{i,j} : i = 1, ..., K_j)$, where $\sum_{i=1}^{K_j} \alpha_{i,j} = 1$. We note that this is *one* possible model of a matching market that is closer in spirit to SSAP [13] as well as other related formulations thereof; it differs from other models in the matching literature (see §1.2) in that it tries to capture a salient aspect of large markets, viz., choice overload, as opposed to traditional aspects such as *competition* and *congestion* best elucidated via conventional "finite" market models.

The platform's goal is to maximize its expected cumulative payoffs over a sequence of $n$ rounds of matching, subject to worker-types w.r.t. job-types and their distributions $\{\boldsymbol{\alpha_j} : j \in \mathcal{J}\}$, as well as mean payoffs for possible worker-job type-pairs being latent attributes. As is the norm in settings with incomplete information and imperfect learning, we reformulate this objective as minimizing the *expected cumulative regret* relative to an oracle that is privy to aforementioned primitives.

**On the complexity of the problem.** Even with a unique job-type, say $\mathcal{J} = \{j_0\}$, and only one job arriving per period, the ensuing *allocation* problem is challenging to analyze on account of the distribution $\boldsymbol{\alpha_{j_0}}$ and any statistical properties of the rewards being unknown. In the simplest possible formulation, $K_{j_0} = 2$, and the statistical complexity of the corresponding regret minimization problem is governed by three principal primitives: (i) the sub-optimality gap $\underline{\Delta}_{j_0} > 0$ between the mean rewards of the optimal and inferior worker sub-populations; (ii) the probability $\alpha_{1,j_0}$ of sampling an optimal worker from the population; and (iii) the planning horizon $n$. One may aptly recognize this as an infinitely many-armed bandit problem (where arms are synonymous to workers) with an *arm-reservoir* distribution $(\alpha_{1,j_0}, 1 - \alpha_{1,j_0})$ and a mean reward gap of $\underline{\Delta}_{j_0}$. However, this model differs from the classical literature on infinite-armed bandits in that its arm-reservoir distribution is not endowed with any regularity properties (see the literature review in §1.2), instead we only posit a finite support with cardinality known to the decision maker (in this case, a cardinality of two), absent however, knowledge of the associated probability masses (in this case, $\alpha_{j_0,1}$ and $1 - \alpha_{j_0,1}$). In our setting, absence of information on $\alpha_{1,j_0}$ significantly exacerbates the difficulty of analysis as calibrating *exploration* becomes challenging (on account of a "large" number of arms). In particular, how many arms must one query from the arm-reservoir in order to have at least one optimal arm in the queried set with high probability, is difficult to answer if (a lower bound on) the proportion $\alpha_{1,j_0}$ of optimal arms is unknown. Consequently, any finite consideration set may only contain inferior arms and as a result, any algorithm limited to such a selection will suffer a *linear regret*. One may contrast this setting with its classical two-armed counterpart with gap $\underline{\Delta}_{j_0}$, where finiteness of the set of arms (binary action space) makes it possible to design efficient rate-optimal policies. In our setting, on the other hand, it remains a priori unclear if there even exists a policy capable of achieving *sub-linear regret*. The general matching problem naturally is only harder.

## 1.1 Contributions

In this work, we resolve several foundational questions pertaining to complexity and achievable performance in the matching problem described earlier, and provide a comprehensive understanding of various other aspects thereof. Among other things, we propose an algorithm that achieves a finite-time instance-dependent expected regret of $\mathcal{O}(\log n)$ after $n$ rounds (the big-Oh subsumes problem-dependent scaling factors encapsulating its fundamental complexity), and prove that this

performance cannot be improved w.r.t. $n$. While the order of regret and complexity of the problem suggests a great degree of similarity to the classical stochastic finite-armed bandit problem, properties of the performance bounds and salient aspects of algorithm design are quite distinct from the latter, as are the key primitives that determine complexity along with the analysis tools needed to study them. In what follows, we will for expositional reasons assume $\mathcal{J} = \{j_0\}$ and a batch size of 1 (jobs arrive one at a time) whenever $|\mathcal{J}| = 1$. Our theoretical contributions can then be projected along the following axes:

**Complexity of regret when $|\mathcal{J}| = 1$.** We establish information-theoretic lower bounds on regret that are order-wise tight (in the horizon $n$) in the instance-dependent setting (Theorem 1). In addition, we establish a *uniform* lower bound on achievable performance (tight in $n$) that captures explicitly the scaling behavior w.r.t. the fraction $\alpha_{1,j_0}$ of optimal arms (Theorem 2); this is shown via a novel non-information-theoretic proof based entirely on convex analysis.

**Algorithm design and achievable performance.** We propose a policy (Algorithm 2) that is rate-optimal (in $n$) for the instance-dependent setting. Our policy relies only on knowledge of $K_{j_0}$, is agnostic to information pertaining to the reward distributions as well as the distribution $(\alpha_{i,j_0} : i = 1, ..., K_{j_0})$ of worker-types. Furthermore, the upper bound depends on the distribution of worker-types only in sub-logarithmic terms (see below).

**Performance bounds for general $\mathcal{J}$.** Aforementioned results for $|\mathcal{J}| = 1$ and unit batch size are then translated to the general (matching) version of the problem described earlier, where $\mathcal{J}$ can be any arbitrary finite set and jobs may arrive in batches of stochastic size and constitution. In the matching problem, we establish that regret is bounded above by $\sum_{j \in \mathcal{J}} (C_1(\boldsymbol{\mu_j}) \log n + C_2(\boldsymbol{\mu_j}, \boldsymbol{\alpha_j}) \log \log n)$ under our policy tailored to this setting, where the constants $C_1(\cdot), C_2(\cdot, \cdot)$ only depend on their arguments, $\boldsymbol{\mu_j} := (\mu_{i,j} : i = 1, ..., K_j)$[1] and $\boldsymbol{\alpha_j} := (\alpha_{i,j} : i = 1, ..., K_j)$ (Theorem 4). When $K_j = 2 \, \forall \, j \in \mathcal{J}$, we improve this guarantee to $\sum_{j \in \mathcal{J}} (C_1(\boldsymbol{\mu_j}) \log n + C_2(\boldsymbol{\mu_j}, \boldsymbol{\alpha_j}))$ (Theorem 6). It is noteworthy that the upper bounds depend on $\{\boldsymbol{\alpha_j} : j \in \mathcal{J}\}$ only in sub-logarithmic terms. We believe this finding may be of interest more broadly.

## 1.2 Literature review

Our work is positioned relative to two major streams of literature; dynamic matching and multi-armed bandits. Below, we briefly survey each of these areas and remark on the distinctions and novelties in our model vis-à-vis the extant body of work.

**Dynamic matching under uncertainty.** There is a recent line of work on simultaneous learning and matching in bipartite graphs under uncertainty. For a survey of works in this area, see [22, 23, 24, 25, 16, 9, 3], etc. Aforementioned references, by and large, consider an archetypal stable matching problem under uncertainty in preferences where a heterogeneous collection of jobs (represented by nodes on one side of a bipartite graph) must be matched to workers (the other side of the graph) with unknown or noisy preferences over jobs. The matching proceeds iteratively in rounds in a way that meets certain stability criteria at all times as well as ensures that the true preferences are "learnt" at a regret-optimal rate. Cited works, however, differ from our paper fundamentally in that their learning problems are posited over a *finite* set of workers, which allows for sufficient exploration of each; this would be infeasible in our setting owing to a "large" population thereof.

In contrast, [17] considers a stationary setting where a stream of heterogeneous jobs must dynamically be matched to a policy-dependent steady state population of workers in a way that respects capacity constraints on the supply and demand processes. This paper shares basic similarities with our work in studying a "large" population model of workers. Their key technical innovation, however, lies in the way polytope capacity constraints are handled via an intelligent use of shadow prices to create essentially an unconstrained learning problem that may be solved rate-optimally using conventional heuristics. It is noteworthy that the algorithm proposed in aforementioned reference requires ex ante knowledge of $\{\boldsymbol{\mu_j}, \boldsymbol{\alpha_j} : j \in \mathcal{J}\}$ in addition to other primitives. Our model, on the other hand, has a richer learning component that is challenging to address as it is, absence of capacity constraints notwithstanding. Our primary contribution here lies in establishing fundamental achievability results for this setting and in the design of novel rate-optimal algorithms that adapt to key primitives online.

---

[1] $\mu_{i,j}$ denotes the mean reward per match between a worker of type $i$ (w.r.t. job-type $j$) and a type $j$ job.

**Multi-armed bandits.** Our problem shares structural similarities with *infinitely many-armed bandits*, modulo heterogeneity and multiplicity of pulls. Infinite-armed bandits involve a fixed *reservoir distribution* over an *uncountable* set of arm-types (possible mean rewards) that may be queried arbitrarily often over the horizon of play. These problems trace their roots to [4] which studied the Bernoulli reward setting under a Uniform (on $[0,1]$) prior on the mean. Subsequent works have considered more general reservoir distributions, albeit endowed with certain *regularity* properties, see, e.g., [26, 6, 8, 10], etc. In terms of the statistical complexity of regret minimization, such regularity assumptions are tantamount to the minimal achievable regret being polynomial in the horizon (see above references). In contrast, our model is fundamentally simpler owing to a finite set of arm-types despite also being endowed with infinitely many arms. However, unlike cited works, the decision maker in our setting is completely oblivious to the reservoir distribution (or any property thereof) which substantially exacerbates the difficulty of analysis as well as the hardness of the problem itself.

### 1.3 Organization of the paper

§2 provides a formal description of the problem and §3 contains results pertaining to lower bounds on achievable performance for natural policy classes. §4 deals with design and analysis considerations for rate-optimal policies and also contains our main propositions together with supporting theoretical guarantees. All other discussion (including auxiliary results and proofs) is deferred to the appendix.

## 2 Problem formulation

**Job-arrival process.** The platform faces an arrival stream of jobs (i.i.d. in time) given by $\{(\Lambda_{j,t} : j \in \mathcal{J}) : t = 1, 2, ...\}$, where $\mathcal{J}$ is finite and $\Lambda_{j,t}$ is the number of type $j$ jobs arriving at time $t$. Types and multiplicities of jobs are perfectly observable upon arrival. We assume that there exists some finite constant $M > 0$ satisfying $\mathbb{P}\left(\max_{j \in \mathcal{J}} \sup_{t \geqslant 1} \Lambda_{j,t} \leqslant M\right) = 1$. We remark that our algorithms do not require knowledge of $M$; the assumption only serves to simplify analysis.[2]

**Supply of workers.** We assume that workers are distributed on the unit interval $[0, 1]$ according to some probability distribution $\mathcal{D}$ that is absolutely continuous w.r.t. the Lebesgue measure on $[0, 1]$. Associated with each job-type $j \in \mathcal{J}$, there exists a permutation $\boldsymbol{\sigma_j} := \{\sigma_j(i) : i = 1, ..., K_j\}$ of $\{1, ..., K_j\}$, and a sequence of thresholds $0 =: \lambda_{0,j} < \lambda_{1,j} < ... < \lambda_{K_j-1,j} < \lambda_{K_j,j} := 1$ partitioning the unit interval into $K_j$ disjoint sub-intervals. We posit a payoff model whereby a worker $x \in (\lambda_{i-1,j}, \lambda_{i,j})$ (for some $i \in \{1, ..., K_j\}$) generates a stochastic reward with mean $\mu_{\sigma_j(i),j}$ upon match with a type $j$ job; it is assumed that the $K_j$ mean rewards adhere to the strict order $\mu_{1,j} > ... > \mu_{K_j,j}$. We define $\alpha_{i,j} := \mathbb{P}\left(X \in \left(\lambda_{\iota(i,j)-1,j}, \lambda_{\iota(i,j),j}\right)\right)$, where $X \sim \mathcal{D}$ and $\iota(i,j) \in \{1, ..., K_j\}$ is the unique element satisfying $\sigma_j\left(\iota(i,j)\right) = i$, as the probability that a worker sampled at random from $\mathcal{D}$ (equivalently, from the *population*), is $i^{\text{th}}$ best for job-type $j$ (generates mean reward $\mu_{i,j}$); such a worker is said to have type $i$ w.r.t. job-type $j$. Thus, a type 1 worker w.r.t. job-type $j$ is *optimal* for jobs of type $j$. Note that the model allows for staggered optimality of worker-types associated with different job-types, as Figure 1 below illustrates.

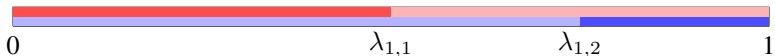

Figure 1: **Possible distribution of worker-types for $\mathcal{J} = \{1, 2\}$ and $K_1 = K_2 = 2$.** The darker shades represent optimal (type 1) workers while the lighter shades represent inferior (type 2) ones w.r.t. each job-type. In this example, no worker can simultaneously be optimal for both job-types.

**High-level description of the matching problem.** Each arriving job may be matched one-to-one to a worker from the available supply. Each match takes at most one period for execution, a matched worker thus frees up before the next lot of jobs arrives. Matched jobs leave the system upon completion and the platform receives a stochastic reward for each completed job; a job that remains unmatched drops out instantaneously. The platform has information neither on individual

---

[2]Though our regret bounds will scale linearly with $M$ as we shall later see, proofs do not as such require batch-sizes to be almost surely bounded and can be refined to guarantee $\mathcal{O}(\log n)$ bounds also for $\Lambda_{j,t}$'s supported on $\mathbb{Z}_+$ under appropriate tail behavior. We do not pursue this line of analysis here purely for expositional reasons.

worker-types w.r.t. job-types nor on their supply distribution, however, it has perfect knowledge of $(K_j : j \in \mathcal{J})$. Subject to this premise, the platform must match incoming jobs to workers in a way that maximizes its expected cumulative payoffs over a sequence of $n$ rounds of matches.

**Adaptive control.** For any job that arrives at time $t$, the platform can match it to: (i) a worker that has matched before, (ii) a *new* worker (one without any history of matches) sampled from the population, or (iii) no worker (job is dropped). A policy $\pi := (\pi_1(\cdot, \cdot), \pi_2(\cdot, \cdot), ...)$ is an *adaptive* rule that prescribes the allocation $\pi_t(\cdot, \cdot)$ at time $t$. Specifically, $\pi_t(j, k)$ denotes the label of the worker (not its *type*) that gets matched to the $k^{\text{th}}$ job of type $j$ arriving at time $t$ (provided there are at least $k$ job-arrivals of type $j$ at $t$ and the $k^{\text{th}}$ job is not dropped). Upon match, a $[0, 1]$-valued stochastic reward with mean $\mu_{\kappa_j(\pi_t(j,k)),j}$ is realized, where $\kappa_j(\pi_t(j, k)) \in \{1, ..., K_j\}$ encodes the type of worker $\pi_t(j, k)$ w.r.t. job-type $j$. The realized rewards are independent across matches and in time.

**Platform's objective.** The goal of maximizing the expected cumulative payoffs over $n$ rounds is converted to minimizing the expected *regret* relative to a clairvoyant policy that prescribes an "optimal" match for each arriving job. We are thus interested in the following optimization problem

$$\inf_{\pi \in \Pi} \mathbb{E} R_n^\pi := \inf_{\pi \in \Pi} \mathbb{E} \left[ \sum_{t=1}^n \sum_{j \in \mathcal{J} : \Lambda_{j,t} \geqslant 1} \sum_{k=1}^{\Lambda_{j,t}} \left( \mu_{1,j} - \mu_{\kappa_j(\pi_t(j,k)),j} \right) \right]. \tag{1}$$

Here, $\Pi$ is the class of *non-anticipating* policies, i.e., $\pi_{t+1}(\cdot, \cdot)$ is *adapted* to $\mathcal{F}_t$ for all $t \in \{0, 1, ...\}$, where $\mathcal{F}_t := \sigma \{(\Lambda_s, \pi_s, r_s) : s = 1, ..., t\}$ denotes the natural filtration at time $t$. Here, $\Lambda_s := (\Lambda_{j,s} : j \in \mathcal{J})$, $\pi_s$ is the set of matches implemented at time $s$ and $r_s$ is the set of collected rewards. The expectation in (1) is w.r.t. the randomness in job-arrivals, worker supply, policy, and rewards.

Going forward, we will adopt standard terminology from the multi-armed bandit literature and refer to workers as "arms" and jobs as "pulls" interchangeably.

## 3 Lower bounds for natural policy classes

For a succinct illustration of the statistical complexity of our problem setting, it is conducive to pivot to the paradigmatic case where the arrival stream comprises only of a single job-type (say $j$) arriving one at a time, and the population of workers is partitioned into $K_j = 2$ clusters with $\alpha_{1,j} \leqslant 1/2$ (recall that this is the proportion of the optimal worker-type). In this case, one anticipates the problem to be at least as hard as the classical two-armed bandit with a mean reward gap of $\underline{\Delta}_j := \mu_{1,j} - \mu_{2,j} > 0$; this is on account of the infinitely many alternatives available to the decision maker in our setting as opposed to just two. Indeed, we establish this in Theorem 1 via an information-theoretic approach that delicately handles the combinatorial complexity arising due to probabilities over countably many arms (proof is provided in Appendix C). In what follows, an instance of the problem refers to a collection of reward distributions with means $(\mu_{1,j}, \mu_{2,j})$ (and gap $\underline{\Delta}_j$). We will overload the notation for expected regret slightly to emphasize its instance-dependence.

**Theorem 1 (Information-theoretic lower bounds on achievable performance)** *Suppose that the arrival process comprises only of a single job-type, say $j \in \mathcal{J}$, arriving one at a time, i.e., $\Lambda_{j,t} = 1$ for $t = 1, 2, ...$ Also suppose that $K_j = 2$ and $\alpha_{1,j} \leqslant 1/2 - \epsilon$, where $\epsilon \in (0, 1/2)$ is arbitrary. Let $\Pi_{adm}$ denote the class of admissible[3] policies. Then, the following is true under any $\pi \in \Pi_{adm}$:*

*(i) For any $\underline{\Delta}_j > 0$, there exists a problem instance $\nu$ such that $\mathbb{E} R_n^\pi(\nu) \geqslant C \log n / \underline{\Delta}_j$ for $n$ large enough[4], where $C$ is some absolute constant.*

*(ii) For any $n \in \mathbb{N}$, there exists a problem instance $\nu$ such that $\mathbb{E} R_n^\pi(\nu) \geqslant \epsilon C' \sqrt{n}$, where $C'$ is some absolute constant.*

**Distinction from classical MAB.** Although the above result bears resemblance to classical information-theoretic lower bounds for finite-armed bandits, it is noteworthy that our setting has a fundamentally greater problem complexity that requires a more nuanced analysis vis-à-vis the finite-armed setting. Traditional proofs, as a result, cannot be translated to our setting in a straight-forward manner. To see this, note that when $\alpha_{1,j}$ is high, a new worker is very likely to be optimal;

---

[3]This is a rich policy class containing all "reasonable" algorithms; refer to Definition (1) in Appendix C.
[4]The dependence on $\epsilon$ is subsumed under "$n$ large enough."

in the limit as $\alpha_{1,j} \to 1$, the problem becomes degenerate as all policies incur zero expected regret. Thus, the problem cannot be harder than the two-armed bandit with gap $\underline{\Delta}_j$ uniformly over all values of $\alpha_{1,j}$. While we conjecture $\alpha_{1,j} < 1$ to be a sufficient condition for the existence of $\Omega\left(\log n/\underline{\Delta}_j\right)$ instance-dependent and $\Omega\left(\sqrt{n}\right)$ instance-independent (minimax) lower bounds (modulo $\alpha_{1,j}$-dependent scaling factors), there are technical challenges due to probabilistic associations over countably many arms. The restriction to $\alpha_{1,j} < 1/2$ and *admissible* policies is then necessary for tractability of the proof and it remains unclear if this can be generalized further.

**Capturing dependence on $\boldsymbol{\alpha_j}$.** Although the instance-dependent lower bound in Theorem 1 is tight in $n$ as we shall later see, it fails to provide actionable insights vis-à-vis $\boldsymbol{\alpha_j}$. A natural question in our setting is whether and in what manner does the presence of countably many arms (as opposed to finitely many) affect achievable regret. In particular, how does the difficulty associated with finding from the available supply an optimal worker for a given job type (and the dependence on the distribution $\boldsymbol{\alpha_j}$) come into play. Below, we propose a lower bound that explicitly captures this dependence, albeit with respect to a somewhat restricted policy class.

**Theorem 2 ($\boldsymbol{\alpha_j}$-dependent lower bound)** *Suppose that the arrival process comprises only of a single job-type, say $j \in \mathcal{J}$, arriving one at a time, i.e., $\Lambda_{j,t} = 1$ for $t = 1, 2, ...$ Also suppose that $\alpha_{1,j} \leqslant 1/2$. Denote by $\Pi_m$ the class of "memoryless" policies under which the decision to match an incoming job to a new worker at any time $t \in \{1, 2, ...\}$ is independent of $\mathcal{F}_{t-1}$. Then, for all problem instances $\nu$ with a minimal sub-optimality gap of at least $\underline{\Delta}_j > 0$, one has*

$$\liminf_{n\to\infty} \inf_{\pi \in \Pi_m} \frac{\mathbb{E} R_n^\pi(\nu)}{\log n} \geqslant \frac{\underline{\Delta}_j}{4\alpha_{1,j}}.$$

**Discussion.** The proof is located in Appendix D. Several comments are in order. **(i)** The class $\Pi_m$, in particular, includes policies that front-load exploration, i.e., sample upfront a pre-specified number of workers from the population and then deploy a regret minimizing algorithm of choice on the set of workers so obtained. This includes several natural approaches to the problem as we shall later see. **(ii)** The foremost noticeable aspect of Theorem 2 that differs from Theorem 1 is that it provides a uniform lower bound *over all instances* that are at least $\underline{\Delta}_j$-separated in the mean reward, as opposed to merely establishing their existence. **(iii)** The presence of $\underline{\Delta}_j$ in the numerator in Theorem 2 unlike traditional information-theoretic bounds where $\underline{\Delta}_j$ resides in the denominator suggests that while this bound may be vacuous if $\underline{\Delta}_j$ is "small," it certainly becomes most relevant when $\underline{\Delta}_j$ is "well-separated." In that sense, Theorem 1 and 2 provide a tool to separate regimes of $\underline{\Delta}_j$ where one bound captures the dominant effects vis-à-vis the other. **(iv)** A novelty of Theorem 2 lies in its proof, which differs from classical lower bound proofs in that it is based entirely on ideas from convex analysis as opposed to the information-theoretic and change-of-measure techniques hitherto used in the literature.

**Remarks**. **(i)** It is not impossible to avoid $1/\alpha_{1,j}$-scaling in the instance-dependent logarithmic regret. We will later show via an upper bound for our algorithm `CAB-K` that the $\alpha_{1,j}$-dependence can, in fact, be relegated to sub-logarithmic terms (`CAB-K` samples new workers from the population *adaptively* based on the sample-history of onboarded workers and therefore does not belong to $\Pi_m$). Importantly, this will establish a somewhat surprising fact that the instance-dependent logarithmic bound in Theorem 1 is optimal w.r.t. to its dependence on $\alpha_{1,j}$ (the scaling w.r.t. $\underline{\Delta}_j$, however, may not be best possible as forthcoming upper bounds will suggest). **(ii)** Theorem 2 holds also for any worker supply where the optimal mean reward w.r.t. job-type $j$ is at least $\underline{\Delta}_j$-separated from the rest, the nature of the set of possible worker-types (countable or uncountable) notwithstanding.

## 4 Designing adaptive policies for matching

The approach we adopt in this paper directly addresses the fact that there is an unlimited supply of available workers at all times. A natural design then is to tailor sub-routines specific to job-types in $\mathcal{J}$ and instantiate them at the first arrival of each type. Specifically, if jobs of type $j$ arrive at $\{t_1, t_2, ...\}$, then the platform should call the sub-routine specific to job-type $j$ only at aforementioned times, independent of other job-arrivals. This leads to the meta-algorithm `MATCH` (see Algorithm 1) for the matching problem. In what follows, `ALG` refers to an arm-allocation rule w.r.t. a fixed job-type that prescribes *one* arm upon each invocation. `ALG` can be thought of as a horizon-free sampling strategy for a countably many-armed bandit problem with one pull per period. When multiple jobs (say $L$) of the same type (say $j$) arrive at the same time, we instantiate (if necessary) new *parallel threads* of

ALG specific to job-type $j$ so there exist a total of $L$ type $j$ threads when the next lot of jobs arrives. In the following, $L_j$ denotes the running count of the number of parallel threads of ALG for type $j$ jobs.

---

**Algorithm 1** MATCH

---
1: **Input:** (i) $\mathcal{J}$, (ii) $(K_j : j \in \mathcal{J})$, and (iii) ALG.
2: **Initialization:** Set $L_j = 0$ for each $j \in \mathcal{J}$.
3: **for** $t \in \{1, 2, ...\}$ **do**
4:     **for** $j \in \mathcal{J}$ **do**
5:         **if** $\Lambda_{j,t} \geqslant 1$ **then**
6:             **if** $\Lambda_{j,t} \leqslant L_j$ **then**
7:                 Match the $\Lambda_{j,t}$ type $j$ jobs to the first $\Lambda_{j,t}$ threads of ALG for type $j$ jobs.
8:             **else**
9:                 Match the first $L_j$ type $j$ jobs to the $L_j$ available threads of ALG for type $j$ jobs.
10:                Instantiate $\Lambda_{j,t} - L_j$ new threads of ALG for the remaining $\Lambda_{j,t} - L_j$ jobs.
11:                Update $L_j \leftarrow \Lambda_{j,t}$.

---

**Discussion of MATCH.** An immediate observation from Algorithm 1 is that ALG must be *anytime*, i.e., it should not depend on the horizon of play since the number of job arrivals (over the platform's planning horizon) of each type is not known a priori. Keeping this objective in mind, we shift our focus to designing an arm-allocation rule ALG w.r.t. a fixed job-type, say type $j$, that: (i) prescribes one pull per period, (ii) depends only on $K_j$, (iii) is adaptive to the mean reward vector $\boldsymbol{\mu_j}$ and the supply distribution $\boldsymbol{\alpha_j}$, and (iv) is horizon-free. Once such an ALG is designed, its composition with MATCH will transfer learning guarantees to the original matching problem.

### 4.1 Shifting focus to adaptive sequential sampling strategies tailored to a specific job-type

Going forward, we will assume that jobs belong to a common fixed type and arrive one at a time. With slight abuse of notation, the supply of available workers is characterized by $K$ worker-types with distinct means $\boldsymbol{\mu} := (\mu_i : i = 1, ..., K)$ adhering to $\mu_1 > ... > \mu_K$. The maximal and minimal sub-optimality gaps are given by $\bar{\Delta} := \mu_1 - \mu_K$ and $\underline{\Delta} := \mu_1 - \mu_2$ respectively, and the minimal reward gap is $\delta := \min_{1 \leqslant i < i' \leqslant K} (\mu_i - \mu_{i'})$. The distribution of worker-types is denoted by $\boldsymbol{\alpha} := (\alpha_i : i = 1, ..., K)$, where $\alpha_i$ is the probability of sampling a type $i$ arm (characterized by mean $\mu_i$) from the population. The learning horizon is $n$. The decision maker only knows $K$ and is oblivious to $(\boldsymbol{\mu}, \boldsymbol{\alpha}, n)$.

The specific setting described above was first studied in [18] for $K = 2$ for which a UCB-styled algorithm with $\mathcal{O}(\log n)$ regret was proposed. The analysis of said algorithm leveraged certain concentration properties of the UCB1 policy [1] that were recently discovered in the context of a two-armed bandit with *equal* arm-means (see Theorem 4(i) in [18]). Currently, there is no known algorithm for the general setting with $K > 2$ arm-types. We discuss in the appendix why properties of UCB1 critical to $\mathcal{O}(\log n)$ regret in [18] do not hold for $K > 2$; consequently, a natural adaptation of their algorithm to $K > 2$ will likely fail to generalize $\mathcal{O}(\log n)$ bounds.[5] To our best knowledge, the general countable-armed bandit (CAB) setting with $K > 2$ types remained open in prior literature.

We close this gap in the literature by proposing a policy CAB-K (see Algorithm 2) that achieves $\mathcal{O}(\log n)$ regret in the general $K$-typed setting. CAB-K operates based on the Explore-then-Commit principle with adaptive stopping and re-initialization times. It is noteworthy that CAB-K is not horizon-free; in particular, knowledge of the horizon is critical for appropriate calibration of its stopping and re-initialization thresholds. However, this is not a constraining characteristic and we will use a doubling trick dubbed HF (Algorithm 3) to make it horizon-free with an anytime $\mathcal{O}(\log n)$ guarantee.

**Discussion of CAB-K.** At any time, the algorithm computes two thresholds of $\mathcal{O}(\sqrt{m \log m})$ and $\mathcal{O}(\sqrt{m \log n})$ for the $\binom{K}{2}$ pairwise-difference-of-reward processes, $m$ being the per-arm sample count. If the envelope of said process is dominated by the former threshold, the concerned pair likely contains arms of the same type (equal means). The explanation stems from the Law of the Iterated Logarithm (see [14], Theorem 8.5.2): the envelope of a zero-drift length-$m$ random walk grows as $\mathcal{O}(\sqrt{m \log \log m})$. In the aforementioned scenario, the algorithm discards the *entire* consideration

---

[5]We will, however, propose a version in the appendix that is asymptotically optimal (achieves $o(n)$ regret).

---

**Algorithm 2** `CAB-K`

---

 1: **Input:** Horizon of play $n$.
 2: Set budget $T = n$.
 3: **Initialize new epoch:** Query $K$ *new* arms; call it consideration set $\mathcal{A} = \{1, 2, ..., K\}$.
 4: Play each arm in $\mathcal{A}$ once; observe rewards $\{X_{a,1} : a \in \mathcal{A}\}$.
 5: Update budget: $T \leftarrow T - K$.
 6: Per-arm sample count $m \leftarrow 1$.
 7: Generate $\binom{K}{2}$ independent standard Gaussian random variables $\{\mathcal{Z}_{a,b} : a, b \in \mathcal{A}, a < b\}$.
 8: **while** $T \geqslant K$ **do**
 9:     **if** $\exists\, a, b \in \mathcal{A},\, a < b$ s.t. $\left|\mathcal{Z}_{a,b} + \sum_{k=1}^{m}\left(X_{a,k} - X_{b,k}\right)\right| < 4\sqrt{m \log m}$ **then**
10:         Permanently discard $\mathcal{A}$ and repeat from step (3).
11:     **else**
12:         **if** $\left|\sum_{k=1}^{m}\left(X_{a,k} - X_{b,k}\right)\right| \geqslant 4\sqrt{m \log n}\; \forall\, a, b \in \mathcal{A},\, a < b$ **then**
13:             Permanently commit to arm $a^* \in \arg\max_{a \in \mathcal{A}}\{\sum_{k=1}^{m} X_{a,k}\}$.
14:         **else**
15:             Play each arm in $\mathcal{A}$ once; observe rewards $(X_{1,m+1}, ..., X_{K,m+1})$.
16:             $m \leftarrow m + 1$.
17:             $T \leftarrow T - K$.

---

set and ushers in a new epoch. This is done to avoid the possibility of incurring linear regret should an optimal arm be missing from the consideration set (e.g., when all arms are type 2). In the other scenario that all pairwise-difference-of-reward processes dominate $\mathcal{O}\left(\sqrt{m \log n}\right)$, the consideration set is likely to contain arms of distinct types (no two have equal means) and the algorithm simply commits to the empirically best arm. Lastly, if neither threshold is crossed (signifying insufficient learning), the sample count for each arm is advanced by one, and the entire process repeats.

**Reason for introducing the Gaussian corruption.** Centered Gaussian noise is added to all pairwise-difference-of-reward processes in step (9) of `CAB-K` to avoid the possibility of incurring linear regret should the support of the reward distributions be a "very small" subset of $[0, 1]$. To illustrate this point, suppose that $K = 2$ and the rewards associated with the types are deterministic with a gap of $\underline{\Delta} < 2\sqrt{2 \log 2}$. Then, as soon as the algorithm queries a consideration set containing one arm each of the two types and the per-arm count reaches 2, the difference-of-reward statistic will satisfy $\left|\sum_{j=1}^{2}\left(X_{1,j} - X_{2,j}\right)\right| = 2\underline{\Delta} < 4\sqrt{2 \log 2}$ and the consideration set will be discarded. On the other hand, if both arms are of the same type (simultaneously optimal or inferior), the algorithm will still re-initialize as soon as the per-arm count reaches 2.[6] This will force the algorithm to keep querying new arms from the reservoir at rate that is linear in time, which is tantamount to incurring linear regret in the horizon. The addition of centered Gaussian noise hedges against this risk by guaranteeing that the difference-of-reward process essentially has an infinite support at all times even when the reward distributions might be degenerate. This rids the regret of its fragility w.r.t. the support of reward distributions. The next proposition crystallizes this discussion.

**Proposition 1 (Persistence of heterogeneous consideration sets)** *Let* $\{X_{a,k} : k = 1, 2, ...\}$ *be a collection of independent samples from an arm of type* $a \in \{1, 2, ..., K\} =: \mathcal{A}$, *and* $\{\mathcal{Z}_{a,b} : a, b \in \mathcal{A}, a < b\}$ *be a collection of* $\binom{K}{2}$ *independently generated standard Gaussians. Then,*

$$\mathbb{P}\left(\bigcap_{m \geqslant 1} \bigcap_{a,b \in \mathcal{A}, a<b} \left\{\left|\mathcal{Z}_{a,b} + \sum_{k=1}^{m}\left(X_{a,k} - X_{b,k}\right)\right| \geqslant 4\sqrt{m \log m}\right\}\right) > \frac{\bar{\Phi}\left(f\left(T_0\right)\right)}{2} =: \beta_{\delta, K} > 0,$$

(2)

*where* $\bar{\Phi}(\cdot)$ *is the tail of the standard Gaussian CDF, and* $T_0 := \max\left(\left\lceil\left(64/\delta^2\right)\log^2\left(\frac{64}{\delta^2}\right)\right\rceil, \mathfrak{C}_K\right)$ *with* $\mathfrak{C}_K := \inf\left\{p \in \mathbb{N} : \sum_{m=p}^{\infty} \frac{1}{m^8} \leqslant \frac{1}{2K^2}\right\}$. *Lastly,* $f(x) := x + 4\sqrt{x \log x}$ *for all* $x \geqslant 1$.

The proof is provided in Appendix E; this meta-result is key to the upper bound stated in Theorem 3.

---

[6] $\left|\sum_{j=1}^{m}\left(X_{1,j} - X_{2,j}\right)\right| = 0$ identically in this case for any $m \in \mathbb{N}$ while $4\sqrt{m \log m} > 0$ only for $m \geqslant 2$.

**Interpretation of $\beta_{\delta,K}$.** First of all, note that $\beta_{\delta,K}$ admits a closed-form characterization in terms of standard functions and satisfies $\beta_{\delta,K} > 0$ for $\delta > 0$ with $\lim_{\delta \to 0} \beta_{\delta,K} = 0$. Secondly, $\beta_{\delta,K}$ *depends exclusively on $\delta$ and $K$*, and represents a lower bound on the probability that CAB-K will never discard a consideration set containing arms of distinct types.

**Theorem 3 (Upper bound on the regret of CAB-K)** *For any horizon of play $n \geqslant 1$, the expected regret of the policy $\pi$ given by CAB-K is bounded as*

$$\mathbb{E} R_n^\pi \leqslant \frac{C K^3 \bar{\Delta}}{\beta_{\delta,K}} \left( \frac{\log n}{\delta^2} + \frac{1}{K! \prod_{i=1}^{K} \alpha_i} \right),$$

*where $\beta_{\delta,K}$ is as defined in (2) and $C$ is some absolute constant.*

**Discussion.** The dependence on the *minimal reward gap $\delta$* is not an artifact of our analysis but, in fact, reflective of the operating principle of the algorithm. CAB-K keeps querying new consideration sets of size $K$ until it determines with high enough confidence that no two arms have the same type (equal means); this is the genesis of $\delta$ in the upper bound. Importantly, equipped just with knowledge of $K$, it remains unclear if there exists an alternative strategy that does not rely on assessing pairwise differences among the queried arms. Another prominent feature of the upper bound is its $\mathcal{O}(1)$-dependence on $\boldsymbol{\alpha}$. This essentially means that the difficulty associated with sampling an optimal arm from an infinite reservoir containing finitely many arm-types is of a *second order*, which starkly contrasts the findings in [12]. Cited paper assumes ex ante knowledge of a lower bound on $\alpha_1$ (proportion of optimal arms) and posits no structure on sub-optimal arm-types. Under this premise, a *first-order* difficulty of sampling optimal arms from the reservoir is established (to wit, the logarithmic term depends on $\alpha_1$). However, whether this would hold also for the subset of problems where the reservoir only contains a finite universe of arm-types (with known cardinality) was left open. Theorem 3 essentially settles this problem. The proof of Theorem 3 is provided in Appendix F.

**Remarks. (i) Possible improvements.** CAB-K, in its present form, indulges in wasteful exploration by discarding entire consideration sets upon re-initialization. It is possible to be parsimonious in this regard and we leave the pursuit of such algorithms to future work. **(ii) More on $\boldsymbol{\beta_{\delta,K}}$.** Notice that when $K = 2$ and $\alpha_1 \leqslant 1/2$, the upper bound in Theorem 3 assumes the form $C \beta_{\delta,2}^{-1} (\log n/\delta + \delta/\alpha_1)$, where $C$ is some absolute constant. Thus, $\beta_{\delta,2}^{-1}$ captures the relative increase in problem complexity (vis-à-vis the paradigmatic two-armed bandit with gap $\delta$) attributable to an unlimited supply of arms of the two types. To what extent may this factor be shaved off remains an interesting open problem. **(iii) Comparison with lower bounds.** One should also contrast Theorem 3 with the lower bound in Theorem 2; by allowing for policies that query the arm-reservoir *adaptively*, we could achieve a regret performance robust to $\boldsymbol{\alpha}$ (second-order dependence). This also leads to the somewhat remarkable conclusion that the lower bound in Theorem 1 is optimal w.r.t. its dependence on $\boldsymbol{\alpha}$. **(iv) Anytime guarantees.** A horizon-free version of CAB-K may be obtained by passing it to the HF operator given in Algorithm 3; a logarithmic bound for the resulting composition HF(CAB-K) is stated in Theorem 8.

### 4.2 Transferring learning guarantees to the matching problem

**Theorem 4 (Achievable performance under MATCH ∘ HF(CAB-K))** *Denote by $\pi$ the composition of MATCH with HF(CAB-K). Then, its expected regret after any number $n \geqslant 1$ of rounds satisfies*

$$\mathbb{E} R_n^\pi \leqslant C M \sum_{j \in \mathcal{J}} \left[ \frac{K_j^3 \bar{\Delta}_j}{\beta_{\delta_j, K_j}} \left( \frac{\log n}{\delta_j^2} + \frac{\log \log (n+2)}{K_j! \prod_{i=1}^{K_j} \alpha_{i,j}} \right) \right],$$

*where $\beta_{\delta_j, K_j}$ is as defined in (2) with $\delta \leftarrow \delta_j := \min_{1 \leqslant i < i' \leqslant K_j} (\mu_{i,j} - \mu_{i',j})$ and $K \leftarrow K_j$, $\bar{\Delta}_j := \mu_{1,j} - \mu_{K_j, j}$, and $C$ is some absolute constant.*

**Discussion.** The foremost noticeable aspect of Theorem 4 is that achievable regret depends on $\{ \boldsymbol{\alpha}_j : j \in \mathcal{J} \}$ (collection of worker-type distributions w.r.t. job-types), surprisingly, only through sub-logarithmic terms. Moreover, when $K_j = 2 \; \forall \, j \in \mathcal{J}$, we improve this to an $\mathcal{O}(1)$-dependence (see Theorem 6 in the appendix). We conjecture that the $\mathcal{O}(\log \log n)$ factor in the upper bound can, in fact, be shaved off also for $K_j > 2$; pursuits in this direction are left to future work. Among other things, characterizing the *minimax* complexity of this setting remains a challenging open problem in light of the multiplicative factors that appear in Theorem 4 (fundamentally different from finite-armed problems). Numerical experiments showing $\mathcal{O}(\log n)$ achievable regret are provided in the appendix.

## Acknowledgments and Disclosure of Funding

The authors thank the anonymous reviewers for their constructive feedback on the initial version of this paper. The authors declare an absence of any competing interests, financial or otherwise.

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
