## Supplementary material: General organization

1. Appendix A contains additional discussion pertaining to the propositions in the main text.
2. Appendix B provides numerical experiments.
3. Appendix C provides the proof of Theorem 1.
4. Appendix D provides the proof of Theorem 2.
5. Appendix E provides the proof of Proposition 1.
6. Appendix F provides the proof of Theorem 3.
7. Appendix G discusses a doubling sub-routine to make algorithms horizon-free.
8. Appendix H provides the proof of Theorem 4.
9. Appendix I discusses a first-order optimal policy for countable-armed bandits with $K$ types.
10. Appendix J discusses auxiliary results pertaining to the aforementioned policy.
11. Appendix K provides more explore-then-commit policies for the countable-armed bandit.

## A    Additional discussion

We next provide a regret upper bound for a countable-armed bandit policy `CAB-K(UCB)` that improves upon the UCB-styled algorithm of [18] by ridding its performance guarantee of a certain fragile assumption pertaining to the reward support. The following result, however, is specific to the case of $K = 2$ arm-types. The policy `CAB-K(UCB)` is stated as Algorithm 4 in Appendix I.

**Theorem 5 (Upper bound on the regret of CAB-K(UCB) when $K = 2$)** *The expected regret of the policy $\pi$ given by* `CAB-2(UCB)` *after any number $n \geqslant 1$ of pulls is bounded as*

$$\mathbb{E}R_n^\pi \leqslant \min\left[\underline{\Delta} n, \frac{C}{\beta_{\underline{\Delta},2}}\left(\frac{\log n}{\underline{\Delta}} + \frac{\underline{\Delta}}{\alpha_1}\right)\right],$$

*where $\beta_{\underline{\Delta},2}$ is as defined in (2) with $\delta \leftarrow \underline{\Delta}$ and $K \leftarrow 2$, and $C$ is some absolute constant.*

The proof is provided in §I.2.

**Limitation of CAB-2(UCB).** The performance stated in Theorem 5 together with its anytime property might appear to give an edge to `CAB-2(UCB)` over `CAB-K`. However, the former is theoretically disadvantaged in that its logarithmic upper bound is currently not amenable to generalization to $K > 2$. The issue traces its roots to the use of `UCB1` as a subroutine; concentration behavior thereof leveraged towards the analysis of `CAB-K(UCB)` when $K = 2$ fails to hold when $K > 2$ rendering proofs intractable.[7] This is illustrated via a simple example with $K = 3$ arm-types discussed below.

**Technical issues with generalizing logarithmic bounds of CAB-2(UCB) to $K > 2$ types.** In the $K = 2$ setting, there are only two possibilities for what a consideration set could be; arms can have means that are either (i) distinct, or (ii) equal. In the former case, an optimal arm is guaranteed to exist in the consideration set and `UCB1` will spend the bulk of its sampling effort on it, which is good for regret performance. In the latter scenario, since arms have equal means, `UCB1` will split samples approximately equally between the two with high probability (see Theorem 4(i) in [18]); subsequently the consideration set will be discarded within a finite number of samples in expectation (see steps (6) and (7) of `CAB-K(UCB)`). Contrast this with an alternative setting with $K = 3$ and mean rewards $\mu_1 > \mu_2 > \mu_3$. In this case, `CAB-K(UCB)` will query consideration sets of size 3. Thus, a query can potentially return one arm with mean $\mu_2$ and two with mean $\mu_3$. Since an optimal arm (mean $\mu_1$) is missing, the algorithm will incur linear regret on this set; it is therefore imperative to discard it at the earliest. Unfortunately though, `UCB1` will invest an overwhelming majority of its sampling effort in the "locally optimal" arm (mean $\mu_2$) and allocate logarithmically fewer samples among the other two. This logarithmic rate of sampling arms with mean $\mu_3$ is proof-inhibiting (vis-à-vis the $K = 2$ case where the rate is linear as previously discussed), making it difficult to theoretically answer if `CAB-K(UCB)` might still be able to discard the arms within, say, logarithmically many pulls in the horizon. This is an open research question and at the moment, strong $\mathcal{O}(\log n)$

---

[7]Concentration behavior à la Theorem 4 of [18] is elucidated further in [19].

performance guarantees are available only for $K = 2$; we could only establish asymptotic-optimality ($o(n)$ regret) for $K > 2$ (see Theorem 9). Among other things that remain, identifying the optimal (instance-dependent) scaling factors w.r.t. $(\boldsymbol{\mu}, \boldsymbol{\alpha})$ and the optimal order of minimax regret would be challenging open directions.

**More on the inverse scaling w.r.t. $\boldsymbol{\beta_{\underline{\Delta},2}}$.** The multiplicative factor $\beta_{\underline{\Delta},2} \leqslant 1$ captures the additional complexity of the problem due to the countable nature of arms. Ideally, one would want the sampling strategy to never discard a consideration set of distinct-typed arms (equivalently, the probability in (2) should always be 1). This, however, is not statistically achievable since the rewards are stochastic and types are unobservable. Some "false positives" will be unavoidable (heterogeneous consideration sets incorrectly labeled homogeneous and therefore discarded) causing the aforementioned probability to be bounded away from 1. However, one can show that this probability is bounded away also from 0 (Proposition 1 formalizes this statement), and $\beta_{\underline{\Delta},2}$ provides one such lower bound when $K = 2$. As to what the tightest possible lower bound is as a function of $\underline{\Delta}$ remains an open problem at the moment. This would also shed light on the additional complexity of the problem attributable to the countable nature of arms vis-à-vis the classical finite-armed problem. In terms of regret guarantees, positivity alone of $\beta_{\underline{\Delta},2}$ suffices to establish instance-dependent rate-optimality (in $n$) of `CAB-K(UCB)` when $K = 2$ (see Theorem 1), notwithstanding the structure of its dependence on $\underline{\Delta}$. The worst-case upper bound, however, is polynomially bounded away from the optimal $\sqrt{n}$ minimax rate owing to the presence of $\beta_{\underline{\Delta},2}$ in the denominator.

**Composing MATCH with CAB-K(UCB) when $\boldsymbol{K_j = 2 \; \forall \; j \in \mathcal{J}}$.** We are now ready to elevate the performance guarantee of `CAB-2(UCB)` to the broader setting of the matching problem where each job-type partitions workers into two sub-populations.

**Theorem 6 (Achievable performance under MATCH ∘ CAB-2(UCB))** *Denote by $\pi$ the composition of `MATCH` with `CAB-2(UCB)`. Then, its expected regret after any number $n \geqslant 1$ of rounds is bounded as*

$$\mathbb{E}R_n^\pi \leqslant \sum_{j \in \mathcal{J}} \left[ \frac{CM}{\beta_{\underline{\Delta}_j,2}} \left( \frac{\log n}{\underline{\Delta}_j} + \frac{\underline{\Delta}_j}{\alpha_{1,j}} \right) \right],$$

*where $C$ is some absolute constant, and $\beta_{\underline{\Delta}_j,2}$ is as defined in (2) with $\delta \leftarrow \underline{\Delta}_j := \mu_{1,j} - \mu_{2,j}$ and $K_j \leftarrow 2$.*

**In conclusion.** We attempted in this paper to address from first principles the problem of dynamic matching in large markets under type uncertainty on the supply side (worker-types) and size uncertainty on the demand size (batch-sizes). Among things that remain, a predominant open question is how can one incorporate capacity constraints on supply and demand processes into this model, e.g., à la [17]. This would lift our setting to model more realistic scenarios where the population of available workers at any time is "large but finite;" we leave investigations in this direction to future work. Another natural modeling extension is one where workers may interact with the platform *strategically* (this could, e.g., reflect via a time-variant and potentially policy-dependent supply distribution); such *endogeneity* will likely force the platform to align its matching policies also with worker *incentives* in order to plug their potential *attrition*. As to what the fundamental limits of learning and achievable regret might be in such a setting remains an interesting open problem to study. There are also a number of interesting questions more directly on the methodological front and related closely to the technical development in this paper; we leave their pursuit to future work.

# B   Numerical experiments

We will evaluate the empirical performance of `CAB-K` (see Algorithm 2) in the countable-armed bandit problem (with one pull per period) characterized by means $\mu_1 > ... > \mu_K$ and a reservoir distribution $\boldsymbol{\alpha} = (\alpha_i : i = 1, ..., K)$. Recall that $\underline{\Delta} = \mu_1 - \mu_2$ and $\delta = \min_{1 \leqslant i < i' \leqslant K} (\mu_i - \mu_{i'})$. We will conduct experiments for $K = 2$ and $K = 3$.

**Experiments.** In what follows, the graphs show the performance of different algorithms simulated on synthetic data. The horizon is capped at $n = 10^5$ for $K = 2$ and at $n = 10^4$ for $K = 3$. Each regret plot is averaged over at least 100 independent experiments (sample-paths). The shaded regions indicate standard 95% confidence intervals. For horizon-dependent algorithms, regret is plotted for

discrete values of the horizon $n$ indicated by "$*$" and interpolated; for anytime algorithms, regret accrued until each $t \in \{1, ..., n\}$ is plotted.

**Baseline policies.** We will benchmark the performance of `CAB-K` against three policies: (i) `Sampling-UCB` [12], (ii) `ETC-`$\infty$`(2)` [18], and (iii) `ETC-RAW` (see Algorithm 6 in Appendix K). The first of these, `Sampling-UCB`, is a UCB-styled policy based on front-loading exploration of *new* arms (Theorem 2 thus applies to this policy). It is, however, noteworthy that `Sampling-UCB` is predicated on ex ante knowledge of (a lower bound on) the probability $\alpha_1$ of sampling an optimal arm from the reservoir; we reemphasize that this is not the setting of interest in our paper. Furthermore, its regret scales as $\tilde{\mathcal{O}}\left(\log n / \left(\alpha_1 \underline{\Delta}\right)\right)$ (up to poly-logarithmic factors in $1/\underline{\Delta}$), which is inferior in terms of its dependence on $\alpha_1$ relative to `CAB-K` and `CAB-2(UCB)` (see Theorem 3 and 5 respectively). There exist other algorithms as well (see, e.g., [11, 27]) developed for formulations with a prohibitively large number of arms. However, these are either sensitive to certain parametric assumptions on the probability of sampling an optimal arm, or focus on a different notion of regret altogether; both directions remain outside the ambit of our setting.

The second policy `ETC-`$\infty$`(2)` is a non-adaptive explore-then-commit-styled algorithm for reservoirs with $K = 2$ types; this policy requires ex ante knowledge of a lower bound on the difference between the two mean rewards. Although `ETC-`$\infty$`(2)` was originally proposed only for $K = 2$, it is easily generalizable and we present in Algorithm 5 (see Appendix K) a version (`ETC-`$\infty$`(K)`) that is adapted to $K$ types.

The last policy `ETC-RAW` is also based on the explore-then-commit principle and operates using a pre-specified exploration schedule as opposed to an adaptive one à la `CAB-K`. We do not provide any theoretical performance guarantees for this policy and resort directly to empirical evaluations.

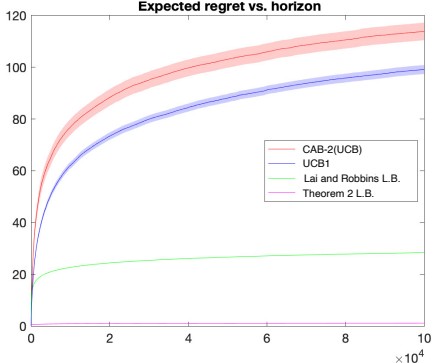

Figure 2: $K = 2$ and $\alpha = (1/2, 1/2)$: Achievable regret in the countable-armed problem vis-à-vis the paradigmatic two-armed bandit.

Figure 3: $K = 2$ and $\alpha = (1/2, 1/2)$: An instance with Bernoulli $0.6, 0.4$ rewards.

**Setup 1 [Figure 2, 3 and 4].** In this setting, we consider $K = 2$ with $\alpha_1 = 0.5$, i.e., two equiprobable arm-types, characterized by Bernoulli$(0.6)$ and Bernoulli$(0.4)$ rewards. Via this setup, we intend to illustrate the difference between the empirical performance achievable in the countable-armed setting vis-à-vis its traditional two-armed counterpart. Refer to Figure 2. The red curve indicates the empirical performance of `CAB-2(UCB)` in this setting. For reference, the blue one shows the empirical performance of `UCB1` [1] in a two-armed bandit with Bernoulli$(0.6)$ and Bernoulli$(0.4)$ rewards; the green curve indicates the best achievable instance-dependent regret [20] in said two-armed configuration. As expected, the regret of `CAB-2(UCB)` is inflated relative to `UCB1`. This is owing to the $\beta_{\delta,2} \leqslant 1$ factor present in the denominator of `CAB-2(UCB)`'s upper bound; characterizing the sharpest lower bound on the probability in (2) (see Proposition 1) is challenging owing to the limited theoretical tools available to this end and we leave it as an open problem at the moment. Figure 3 shows the empirical performance of the algorithms discussed previously as well as `Sampling-UCB` initialized with $\alpha_1 = 1/2$ and `ETC-`$\infty$`(2)` initialized with $\underline{\delta} = \delta/2 = 0.1$. Evidently, the (adaptive) explore-then-commit approach in `CAB-K` outperforms the pre-specified exploration schedule-based approach of `ETC-RAW`, and performs almost as good as the *gap-aware* approach in `ETC-`$\infty$`(2)`. While `Sampling-UCB` outmatches all explore-then-commit styled approaches, the best performing algorithm

is `CAB-K(UCB)`. Surprisingly, this is despite the fact that the theoretical performance bounds for `CAB-K` and `CAB-K(UCB)` are identical (modulo numerical multiplicative constants) when $K = 2$ and $\alpha_1 \leqslant 0.5$ (see Theorem 3 and 5). A similar hierarchy in performances is also observable in Figure 4, which corresponds to a slightly "easier" instance with $\delta = 0.4$ (as opposed to $0.2$) and equiprobable Bernoulli$(0.9)$ and Bernoulli$(0.5)$ rewards.

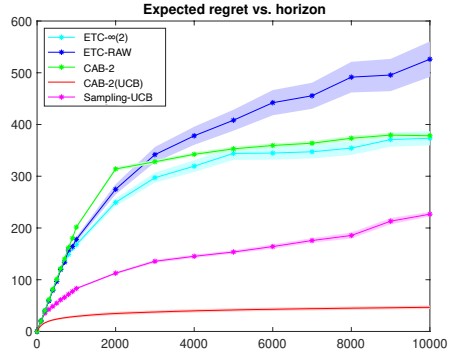
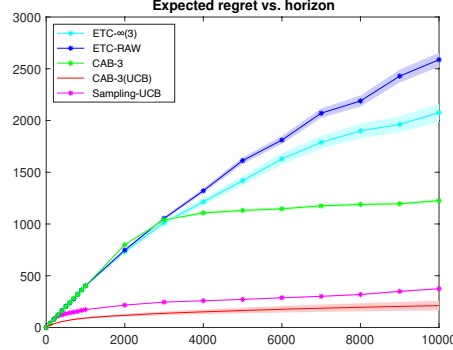

Figure 4: $K = 2$ and $\alpha = (1/2, 1/2)$: An instance with Bernoulli $0.9, 0.5$ rewards.

Figure 5: $K = 3$ and $\alpha = (1/3, 1/3, 1/3)$: An instance with Bernoulli $0.9, 0.5, 0.1$ rewards.

**Setup 2 [Figure 5].** Here, we consider a setting with $K = 3$ arm-types characterized by Bernoulli rewards with means $0.9, 0.5, 0.1$, each occurring with probability $1/3$. We compare the performance of `ETC-RAW`, `CAB-K` and `CAB-K(UCB)` with `ETC-∞(3)` initialized with $\underline{\delta} = \delta/2 = 0.2$, and `Sampling-UCB` initialized with $\alpha_1 = 1/3$. It is noteworthy that despite `CAB-K(UCB)`'s significantly superior empirical performance relative to aforementioned algorithms, only a weak $o(n)$ bound on its regret is currently available (see Theorem 9) due to reasons discussed earlier in the paper. Investigating best achievable rates under `CAB-K(UCB)` is an area of active research at the moment.

**Remark.** For general $\mathcal{J}$, our approach to the matching problem (see `MATCH`) involves instantiating horizon-free versions of aforementioned algorithms independently for each job-type upon its first arrival. We have not included in the current version of our paper numerical experiments pertaining to general $\mathcal{J}$ since the complexity of our problem setting, by and large, is encapsulated by $|\mathcal{J}| = 1$ (owing to the design of `MATCH`), which is already discussed above. We do, however, plan to include comprehensive simulations for the general matching setting as well in future iterations of this paper.

## C  Proof of Theorem 1

**Notation.** Since the job-type is fixed (at $j$), we will with slight abuse of notation drop the subscript $j$ from $\left(\mu_{1,j}, \mu_{2,j}, \underline{\Delta}_j, \alpha_{1,j}, \kappa_j(\cdot)\right)$. Also, we will denote by $\pi_t$ the index of the arm played by policy $\pi$ at time $t$ and by $\kappa(\pi_t) \in \{1, 2\}$ its type. Let $\mathcal{G}_1(x)$ and $\mathcal{G}_2(x)$ be two arbitrary collections of distributions with mean $x \in \mathbb{R}$. The tuple $(\mathcal{G}_1(x), \mathcal{G}_2(y))$ will be referred to as an *instance*.

Since the horizon of play is fixed at $n$, the decision maker may play at most $n$ distinct arms. Therefore, it suffices to focus only on the sequence of the first $n$ arms that may be played. A *realization* of an instance $\nu = (\mathcal{G}_1(\bar{\mu}_1), \mathcal{G}_2(\bar{\mu}_2))$ is defined as the $n$-tuple $r \equiv (r_i)_{1 \leqslant i \leqslant n}$, where $r_i \in \mathcal{G}_1(\bar{\mu}_1) \cup \mathcal{G}_2(\bar{\mu}_2)$ indicates the reward distribution of arm $i \in \{1, 2, ..., n\}$. It must be noted that the decision maker need not play every arm in $r$. Let $i^* := \arg\max_{i \in \{1,2\}} \bar{\mu}_i$. Then, the distribution over the possible realizations of $\nu = (\mathcal{G}_1(\bar{\mu}_1), \mathcal{G}_2(\bar{\mu}_2))$ in $\{r : r_i \in \mathcal{G}_1(\bar{\mu}_1) \cup \mathcal{G}_2(\bar{\mu}_2), \ 1 \leqslant i \leqslant n\}$ satisfies $\mathbb{P}\left(r_i \in \mathcal{G}_{i^*}(\bar{\mu}_{i^*})\right) = \alpha_1$ for all $i \in \{1, ..., n\}$.

Recall that the cumulative pseudo-regret after $n$ plays of a policy $\pi$ on $\nu = (\mathcal{G}_1(\bar{\mu}_1), \mathcal{G}_2(\bar{\mu}_2))$ is given by $R_n^\pi(\nu) = \sum_{m=1}^n \left(\bar{\mu}_{i^*} - \bar{\mu}_{\kappa(\pi_t)}\right)$, where $\kappa(\pi_t) \in \{1, 2\}$ indicates the type of the arm played by $\pi$ at time $t$. Our goal is to lower bound $\mathbb{E}R_n^\pi(\nu)$, where the expectation is w.r.t. the randomness in $\pi$ as well as the distribution over the possible realizations of $\nu$. To this end, we define the notion of

expected cumulative regret of $\pi$ on a realization $r$ of $\nu = (\mathcal{G}_1(\bar{\mu}_1), \mathcal{G}_2(\bar{\mu}_2))$ by

$$S_n^\pi(\nu, r) := \mathbb{E}^\pi \left[ \sum_{t=1}^n \left( \bar{\mu}_{i^*} - \bar{\mu}_{\kappa(\pi_t)} \right) \right],$$

where the expectation $\mathbb{E}^\pi$ is w.r.t. the randomness in $\pi$. Note that $\mathbb{E}R_n^\pi(\nu) = \mathbb{E}^\nu S_n^\pi(\nu, r)$, where the expectation $\mathbb{E}^\nu$ is w.r.t. the distribution over the possible realizations of $\nu$. We define our problem class $\mathcal{N}_{\underline{\Delta}}$ as the collection of $\underline{\Delta}$-separated instances given by

$$\mathcal{N}_{\underline{\Delta}} := \left\{ (\mathcal{G}_1(\bar{\mu}_1), \mathcal{G}_2(\bar{\mu}_2)) : \bar{\mu}_1 - \bar{\mu}_2 = \underline{\Delta}, \ (\bar{\mu}_1, \bar{\mu}_2) \in \mathbb{R}^2 \right\}.$$

**Definition 1 (Admissible policy)** *A policy $\pi$ is deemed admissible for the problem class $\mathcal{N}_{\underline{\Delta}}$ if for any instance $\nu \in \mathcal{N}_{\underline{\Delta}}$ and any realization $r$ thereof, it satisfies*

$$\mathbb{E}^\nu \left[ S_n^\pi(\nu, r) \mid \mathcal{L}(r) = m \right] \geqslant \mathbb{E}^\nu \left[ S_n^\pi(\nu, r) \mid \mathcal{L}(r) = k \right] \ \forall \ (m, n, k) : 0 \leqslant m \leqslant k \leqslant n, \quad (3)$$

*where $\mathcal{L}(r)$ denotes the number of "optimal" arms in realization $r$, i.e., arms with mean $\bar{\mu}_{i^*}$.*

The set of such policies is denoted by $\Pi_{\mathrm{adm}}(\mathcal{N}_{\underline{\Delta}})$. We remark that the condition in (3) is not restrictive since it is only natural that any reasonable policy should incur a larger cumulative regret (in expectation) on realizations with fewer optimal arms.

Fix an arbitrary $\underline{\Delta} > 0$ and consider an instance $\nu = (\{Q_1\}, \{Q_2\}) \in \mathcal{N}_{\underline{\Delta}}$, where $Q_1$ and $Q_2$ are unit-variance Gaussian distributions with means $\mu_1$ and $\mu_2$ respectively. Consider an arbitrary realization $r \in \{Q_1, Q_2\}^n$ of $\nu$ and let $\mathcal{I} \subseteq \{1, 2, ..., n\}$ denote the set of inferior arms in $r$ (arms with reward distribution $Q_2$). Consider another instance $\nu' \in \mathcal{N}_{\underline{\Delta}}$ given by $\nu' = \left( \{\widetilde{Q}_1\}, \{Q_1\} \right)$, where $\widetilde{Q}_1$ is another unit variance Gaussian with mean $\mu_1 + \underline{\Delta}$. Now consider a realization $r' \in \{\widetilde{Q}_1, Q_1\}^n$ of $\nu'$ that is such that the arms at positions in $\mathcal{I}$ have distribution $\widetilde{Q}_1$ while those at positions in $\{1, 2, ..., n\} \backslash \mathcal{I}$ have distribution $Q_1$. Notice that $\mathcal{I}$ is the set of optimal arms in $r'$ (arms with reward distribution $\widetilde{Q}_1$), implying $\mathcal{L}(r') = |\mathcal{I}|$. Then, the following always holds:

$$S_n^\pi(\nu, r) + S_n^\pi(\nu', r') \geqslant \left( \frac{\underline{\Delta} n}{2} \right) \left( \mathbb{P}_{\nu, r}^\pi \left( \sum_{i \in \mathcal{I}} N_i(n) > \frac{n}{2} \right) + \mathbb{P}_{\nu', r'}^\pi \left( \sum_{i \in \mathcal{I}} N_i(n) \leqslant \frac{n}{2} \right) \right),$$

where $\mathbb{P}_{\nu, r}^\pi(\cdot)$ and $\mathbb{P}_{\nu', r'}^\pi(\cdot)$ denote the probability measures w.r.t. the instance-realization pairs $(\nu, r)$ and $(\nu', r')$ respectively, and $N_i(n)$ denotes the number of plays up to and including time $n$ of arm $i \in \{1, 2, ..., n\}$. Using the Bretagnolle-Huber inequality (Theorem 14.2 of [21]), we obtain

$$S_n^\pi(\nu, r) + S_n^\pi(\nu', r') \geqslant \left( \frac{\underline{\Delta} n}{4} \right) \exp \left( -D_{\mathsf{KL}} \left( \mathbb{P}_{\nu, r}^\pi, \mathbb{P}_{\nu', r'}^\pi \right) \right),$$

where $D_{\mathsf{KL}} \left( \mathbb{P}_{\nu, r}^\pi, \mathbb{P}_{\nu', r'}^\pi \right)$ denotes the KL-Divergence between $\mathbb{P}_{\nu, r}^\pi$ and $\mathbb{P}_{\nu', r'}^\pi$. Using Divergence decomposition (Lemma 15.1 of [21]), we further obtain

$$S_n^\pi(\nu, r) + S_n^\pi(\nu', r') \geqslant \left( \frac{\underline{\Delta} n}{4} \right) \exp \left( - \left( \frac{D_{\mathsf{KL}} \left( Q_2, \widetilde{Q}_1 \right)}{\underline{\Delta}} \right) S_n^\pi(\nu, r) \right) = \left( \frac{\underline{\Delta} n}{4} \right) \exp \left( -2\underline{\Delta} S_n^\pi(\nu, r) \right),$$

where the equality follows since $\widetilde{Q}_1$ and $Q_2$ are unit variance Gaussian distributions with means separated by $2\underline{\Delta}$. Next, taking the expectation $\mathbb{E}^\nu$ on both the sides above and a direct application of Jensen's inequality thereafter yields

$$\mathbb{E}R_n^\pi(\nu) + \mathbb{E}^\nu S_n^\pi(\nu', r') \geqslant \left( \frac{\underline{\Delta} n}{4} \right) \exp \left( -2\underline{\Delta} \mathbb{E}R_n^\pi(\nu) \right). \quad (4)$$

Consider the $\mathbb{E}^\nu S_n^\pi(\nu', r')$ term in (4). Using a simple change-of-measure argument, we obtain

$$\mathbb{E}^\nu S_n^\pi(\nu', r') = \mathbb{E}^{\nu'} \left[ S_n^\pi(\nu', r') \left( \frac{1 - \alpha_1}{\alpha_1} \right)^{2\left( \mathcal{L}(r') - n/2 \right)} \right]$$

$$\leqslant \mathbb{E}R_n^\pi(\nu') + \mathbb{E}^{\nu'} \left[ S_n^\pi(\nu', r') \left( \frac{1 - \alpha_1}{\alpha_1} \right)^{2\left( \mathcal{L}(r') - n/2 \right)} \mathbb{1} \left\{ \mathcal{L}(r') > n/2 \right\} \right], \quad (5)$$

where the inequality follows since $\alpha_1 \leqslant 1/2$. Now consider the second term on the RHS in (5). It follows that

$$\mathbb{E}^{\nu'}\left[S_n^\pi(\nu',r')\left(\frac{1-\alpha_1}{\alpha_1}\right)^{2\left(\mathcal{L}(r')-n/2\right)}\mathbb{1}\left\{\mathcal{L}(r')>n/2\right\}\right]$$

$$= \sum_{k>n/2}\mathbb{E}^{\nu'}\left[S_n^\pi(\nu',r')\left(\frac{1-\alpha_1}{\alpha_1}\right)^{2\left(\mathcal{L}(r')-n/2\right)}\mathbb{1}\left\{\mathcal{L}(r')=k\right\}\right]$$

$$= \sum_{k>n/2}\left(\frac{1-\alpha_1}{\alpha_1}\right)^{(2k-n)}\mathbb{E}^{\nu'}\left[S_n^\pi(\nu',r')\mathbb{1}\left\{\mathcal{L}(r')=k\right\}\right]$$

$$= \sum_{k>n/2}\left(\frac{1-\alpha_1}{\alpha_1}\right)^{(2k-n)}\mathbb{E}^{\nu'}\left[S_n^\pi(\nu',r')|\,\mathcal{L}(r')=k\right]\mathbb{P}_{\nu'}\left(\mathcal{L}(r')=k\right)$$

$$= \sum_{k>n/2}\left(\frac{1-\alpha_1}{\alpha_1}\right)^{(2k-n)}\mathbb{E}^{\nu'}\left[S_n^\pi(\nu',r')|\,\mathcal{L}(r')=k\right]\binom{n}{k}\alpha_1^k(1-\alpha_1)^{(n-k)}$$

$$= \alpha_1^n\sum_{k>n/2}\binom{n}{k}\left(\frac{1-\alpha_1}{\alpha_1}\right)^k\mathbb{E}^{\nu'}\left[S_n^\pi(\nu',r')|\,\mathcal{L}(r')=k\right]. \tag{6}$$

Recall that $\nu' \in \mathcal{N}_{\underline{\Delta}}$ and $\pi \in \Pi_{\texttt{adm}}\left(\mathcal{N}_{\underline{\Delta}}\right)$. We have

$$\mathbb{E}R_n^\pi(\nu') = \mathbb{E}^{\nu'}S_n^\pi(\nu',r')$$

$$\geqslant \sum_{m=1}^k\mathbb{E}^{\nu'}\left[S_n^\pi(\nu',r')|\,\mathcal{L}(r')=m\right]\mathbb{P}_{\nu'}\left(\mathcal{L}(r')=m\right) \qquad \text{(for any } k\leqslant n)$$

$$\geqslant \mathbb{E}^{\nu'}\left[S_n^\pi(\nu',r')|\,\mathcal{L}(r')=k\right]\mathbb{P}_{\nu'}\left(\mathcal{L}(r')\leqslant k\right). \qquad \text{(using (3))} \tag{7}$$

Note that under the measure $\mathbb{P}_{\nu'}(\cdot)$, the distribution of $\mathcal{L}(r')$ is Binomial$(n,\alpha_1)$. Therefore, using Markov's inequality, $\mathbb{P}_{\nu'}\left(\mathcal{L}(r')>k\right) \leqslant n\alpha_1/k < 2\alpha_1$ for $k > n/2$ (Note that this bound is non-vacuous since the inequality is strict and $\alpha_1 < 1/2$ by assumption). One then has that $\mathbb{P}_{\nu'}\left(\mathcal{L}(r')\leqslant k\right) > 1-2\alpha_1$ for $k > n/2$. Using this observation in (7), we conclude for $k > n/2$ that

$$\mathbb{E}^{\nu'}\left[S_n^\pi(\nu',r')|\,\mathcal{L}(r')=k\right] < \frac{\mathbb{E}R_n^\pi(\nu')}{1-2\alpha_1}. \tag{8}$$

Combining (6) and (8), one obtains

$$\mathbb{E}^{\nu'}\left[S_n^\pi(\nu',r')\left(\frac{1-\alpha_1}{\alpha_1}\right)^{2\left(\mathcal{L}(r')-n/2\right)}\mathbb{1}\left\{\mathcal{L}(r')>n/2\right\}\right] < \frac{\mathbb{E}R_n^\pi(\nu')}{1-2\alpha_1}. \tag{9}$$

Using (5) and (9), one then concludes

$$\mathbb{E}^\nu S_n^\pi(\nu',r') \leqslant \mathbb{E}R_n^\pi(\nu')\left(1+\frac{1}{1-2\alpha_1}\right) \leqslant \frac{2\mathbb{E}R_n^\pi(\nu')}{1-2\alpha_1}. \tag{10}$$

Finally, from (4) and (10), we have

$$\mathbb{E}R_n^\pi(\nu) + \left(\frac{2}{1-2\alpha_1}\right)\mathbb{E}R_n^\pi(\nu') \geqslant \frac{\underline{\Delta}n}{4}\exp\left(-2\underline{\Delta}\mathbb{E}R_n^\pi(\nu)\right)$$

$$\implies \mathbb{E}R_n^\pi(\nu) + \mathbb{E}R_n^\pi(\nu') \geqslant \frac{(1-2\alpha_1)\underline{\Delta}n}{8}\exp\left(-2\underline{\Delta}\mathbb{E}R_n^\pi(\nu)\right)$$

$$\implies \tilde{R}_n \geqslant \frac{(1-2\alpha_1)\underline{\Delta}n}{16}\exp\left(-2\underline{\Delta}\tilde{R}_n\right)$$

$$\implies \tilde{R}_n \geqslant \frac{\epsilon\underline{\Delta}n}{8}\exp\left(-2\underline{\Delta}\tilde{R}_n\right), \tag{11}$$

where $\tilde{R}_n := \max\left(\mathbb{E}R_n^\pi(\nu), \mathbb{E}R_n^\pi(\nu')\right)$.

**Instance-dependent lower bound**

The assertion of the theorem follows from the fact that the inequality (11) is fulfilled only if for any $\varepsilon \in (0,1)$, $\tilde{R}_n$ satisfies for all $n$ large enough $\tilde{R}_n \geqslant (1-\varepsilon)\log n/(2\underline{\Delta})$.

**Instance-independent (minimax) lower bound**

Since $\tilde{R}_n \leqslant \underline{\Delta}n$, it follows from (11) that

$$\tilde{R}_n \geqslant \frac{\epsilon\underline{\Delta}n}{16}\exp\left(-2\underline{\Delta}^2 n\right).$$

Setting $\underline{\Delta} = 1/\sqrt{n}$ now proves the stated assertion. $\qquad\square$

# D  Proof of Theorem 2

**Notation.** Again, since the job-type is fixed (at $j$), we will with slight abuse of notation drop the subscript $(j)$ from $\left(K_j, \underline{\Delta}_j, \alpha_{1,j}\right)$. Also, note that the result is stated for the general setting with $K \geqslant 2$ arm-types where $\underline{\Delta}$ denotes the minimal sub-optimality gap.

Consider now an arbitrary policy $\pi \in \Pi_{\mathfrak{m}}$. Denote by $A_n^\pi$ the number of *distinct* arms played by $\pi$ until time $n$. Consider an arbitrary $k \in \{1, 2, ..., n\}$. Then, conditioned on $A_n^\pi = k$, the expected cumulative regret incurred by $\pi$ is at least

$$\mathbb{E}\left[R_n^\pi|A_n^\pi = k\right] \geqslant (1-\alpha_1)\underline{\Delta}k + (1-\alpha_1)^k\underline{\Delta}(n-k) =: f(k). \tag{12}$$

**Intuition behind** (12). Each of the $k$ arms played during the horizon has at least one pull associated with it. Consider a clairvoyant policy coupled to $\pi$ that learns the best among the $A_n^\pi$ arms played by $\pi$ as soon as each has been pulled exactly once, i.e., after a total of $A_n^\pi$ pulls. Clearly, the regret incurred by said clairvoyant policy lower bounds $\mathbb{E}R_n^\pi$. Further, since $A_n^\pi$ is independent of the sample-history of arms, it follows that the $A_n^\pi$ arms are statistically identical. Thus, conditioned on $A_n^\pi = k$, the expected regret from the first $k$ pulls of the clairvoyant policy is at least $(1-\alpha_1)\underline{\Delta}k$. Also, the probability that each of the $k$ arms is inferior-typed is $(1-\alpha_1)^k$; the clairvoyant policy thus incurs a regret of at least $(1-\alpha_1)^k\underline{\Delta}(n-k)$ going forward. This explains the lower bound in (12). Therefore, for any $k \in \{1, 2, ..., n\}$, we have

$$\mathbb{E}\left[R_n^\pi|A_n^\pi = k\right] \geqslant \min_{k \in \{1,2,...,n\}} f(k) \geqslant \min_{x \in [0,n]} f(x).$$

We will show that $f(x)$ is strictly convex over $[0,n]$ with $f'(0) < 0$ and $f'(n) > 0$. Then, it would follow that $f(\cdot)$ admits a unique minimizer $x_n^* \in (0,n)$ given by the solution to $f'(x) = 0$. The minimum $f(x_n^*)$ will turn out to be logarithmic in $n$. Observe that

$$f'(x) = (1-\alpha_1)\underline{\Delta} + (1-\alpha_1)^x\underline{\Delta}\left[(n-x)\log(1-\alpha_1) - 1\right],$$
$$f''(x) = -(1-\alpha_1)^x\underline{\Delta}\left[2 - (n-x)\log(1-\alpha_1)\right]\log(1-\alpha_1).$$

Since $\underline{\Delta} > 0$, it follows that $f''(x) > 0$ over $[0,n]$. Further, note that

$$f'(0) = -\alpha_1\underline{\Delta} + \underline{\Delta}n\log(1-\alpha_1) < 0,$$
$$f'(n) = (1-\alpha_1)\underline{\Delta} - (1-\alpha_1)^n\underline{\Delta} > 0.$$

Solving $f'(x_n^*) = 0$ for the unique minimizer $x_n^*$, we obtain

$$\left(\frac{1}{1-\alpha_1}\right)^{x_n^*-1} - 1 = (n-x_n^*)\log\left(\frac{1}{1-\alpha_1}\right)$$
$$\implies \left(\frac{1}{1-\alpha_1}\right)^{x_n^*} + x_n^*\log\left(\frac{1}{1-\alpha_1}\right) > n\log\left(\frac{1}{1-\alpha_1}\right)$$
$$\implies 2\left(\frac{1}{1-\alpha_1}\right)^{x_n^*} > n\log\left(\frac{1}{1-\alpha_1}\right),$$

where the last inequality follows using $y > \log y$. Therefore, we have

$$\left(\frac{1}{1-\alpha_1}\right)^{x_n^*} > \frac{n}{2}\log\left(\frac{1}{1-\alpha_1}\right)$$

$$\implies x_n^* > \frac{\log n + \log\log\left(\frac{1}{1-\alpha_1}\right) - \log 2}{\log\left(\frac{1}{1-\alpha_1}\right)}.$$

Thus, for any $k \in \{1, 2, ..., n\}$,

$$\mathbb{E}\left[R_n^\pi | A_n^\pi = k\right] \geqslant f(x_n^*) > (1-\alpha_1)\underline{\Delta}x_n^* > (1-\alpha_1)\left(\frac{\log n + \log\log\left(\frac{1}{1-\alpha_1}\right) - \log 2}{\log\left(\frac{1}{1-\alpha_1}\right)}\right)\underline{\Delta}$$

$$\implies \mathbb{E}R_n^\pi \geqslant (1-\alpha_1)\left(\frac{\log n + \log\log\left(\frac{1}{1-\alpha_1}\right) - \log 2}{\log\left(\frac{1}{1-\alpha_1}\right)}\right)\underline{\Delta}$$

$$\implies \inf_{\pi\in\Pi_{\mathrm{m}}}\frac{\mathbb{E}R_n^\pi}{\log n} \geqslant (1-\alpha_1)\left(\frac{1}{\log\left(\frac{1}{1-\alpha_1}\right)} + \frac{\log\log\left(\frac{1}{1-\alpha_1}\right) - \log 2}{(\log n)\log\left(\frac{1}{1-\alpha_1}\right)}\right)\underline{\Delta}$$

$$\implies \inf_{\pi\in\Pi_{\mathrm{m}}}\frac{\mathbb{E}R_n^\pi}{\log n} \underset{(\dagger)}{\geqslant} (1-\alpha_1)\left(\frac{1-\alpha_1}{\alpha_1} + \frac{\log\log\left(\frac{1}{1-\alpha_1}\right) - \log 2}{(\log n)\log\left(\frac{1}{1-\alpha_1}\right)}\right)\underline{\Delta}$$

$$\implies \inf_{\pi\in\Pi_{\mathrm{m}}}\frac{\mathbb{E}R_n^\pi}{\log n} \geqslant \frac{(1-\alpha_1)^2\underline{\Delta}}{\alpha_1} + (1-\alpha_1)\left(\frac{\log\log\left(\frac{1}{1-\alpha_1}\right) - \log 2}{(\log n)\log\left(\frac{1}{1-\alpha_1}\right)}\right)\underline{\Delta}$$

$$\implies \inf_{\pi\in\Pi_{\mathrm{m}}}\frac{\mathbb{E}R_n^\pi}{\log n} \underset{(\ddagger)}{\geqslant} \frac{\underline{\Delta}}{4\alpha_1} + (1-\alpha_1)\left(\frac{\log\log\left(\frac{1}{1-\alpha_1}\right) - \log 2}{(\log n)\log\left(\frac{1}{1-\alpha_1}\right)}\right)\underline{\Delta},$$

where $(\dagger)$ follows using $\log y \leqslant y - 1$, and $(\ddagger)$ since $\alpha_1 \leqslant 1/2$. Taking the appropriate limit now proves the assertion. $\qquad\square$

## E  Proof of Proposition 1

Consider the following stopping time:

$$\tau := \inf\left\{m \in \mathbb{N} : \exists a, b \in \mathcal{A}, a < b \text{ s.t. } \mathcal{Z}_{a,b} + \sum_{j=1}^m (X_{a,j} - X_{b,j}) < 4\sqrt{m\log m}\right\}.$$

Since $\mathbb{P}\left(\bigcap_{m\geqslant 1}\bigcap_{a,b\in\mathcal{A},a<b}\left\{\left|\mathcal{Z}_{a,b} + \sum_{j=1}^m (X_{a,j} - X_{b,j})\right| \geqslant 4\sqrt{m\log m}\right\}\right) \geqslant \mathbb{P}(\tau = \infty)$, it suffices to show that $\mathbb{P}(\tau = \infty)$ is bounded away from 0. To this end, define the following entities:

$$\mathfrak{C}_K := \inf\left\{p \in \mathbb{N} : \sum_{m=p}^\infty \frac{1}{m^8} \leqslant \frac{1}{2K^2}\right\},$$

$$T_0 := \max\left(\left\lceil\left(\frac{64}{\delta^2}\right)\log^2\left(\frac{64}{\delta^2}\right)\right\rceil, \Lambda_K\right),$$

$$f(x) := x + 4\sqrt{x\log x} \qquad\qquad \text{for } x \geqslant 1.$$

**Lemma 1** *For any $a, b \in \mathcal{A}$ s.t. $a < b$, it is the case that*

$$\{\mathcal{Z}_{a,b} > f(T_0)\} \subseteq \bigcap_{m=1}^{T_0}\left\{\mathcal{Z} + \sum_{j=1}^m (X_{a,j} - X_{b,j}) \geqslant 4\sqrt{m\log m}\right\}.$$

**Proof of Lemma 1.** Note that

$$\mathcal{Z}_{a,b} > f(T_0)$$
$$= T_0 + 4\sqrt{T_0 \log T_0}$$
$$\geqslant m + 4\sqrt{m \log m} \; \forall \; 1 \leqslant m \leqslant T_0$$
$$\underset{(\mathfrak{a})}{\geqslant} \sum_{j=1}^{m} (X_{b,j} - X_{a,j}) + 4\sqrt{m \log m} \; \forall \; 1 \leqslant m \leqslant T_0$$

$$\implies \mathcal{Z}_{a,b} + \sum_{j=1}^{m} (X_{a,j} - X_{b,j}) \geqslant 4\sqrt{m \log m} \; \forall \; 1 \leqslant m \leqslant T_0,$$

where $(\mathfrak{a})$ follows since the rewards are bounded in $[0,1]$, i.e., $|X_{a,j} - X_{b,j}| \leqslant 1$. $\qquad\square$

**Lemma 2** *For $m \geqslant T_0$, it is the case that*

$$\delta \geqslant 8\sqrt{\frac{\log m}{m}}.$$

**Proof of Lemma 2.** First of all, note that $T_0 \geqslant (64/\delta^2) \log^2 (64/\delta^2) \geqslant 64$ (since $\delta \leqslant 1$). For $s = (64/\delta^2) \log^2 (64/\delta^2)$, one has

$$\delta^2 = \frac{64 \log^2 \left(\frac{64}{\delta^2}\right)}{s} \underset{(\mathfrak{b})}{\geqslant} \frac{64 \left[\log \left(\frac{64}{\delta^2}\right) + 2 \log \log \left(\frac{64}{\delta^2}\right)\right]}{s} = \frac{64 \log s}{s},$$

where $(\mathfrak{b})$ follows since the function $g(x) := x^2 - x - 2 \log x$ is monotone increasing for $x \geqslant \log 64$ (think of $\log (64/\delta^2)$ as $x$), and therefore attains its minimum at $x = \log 64$; one can verify that this minimum is strictly positive. Furthermore, since $\log s / s$ is monotone decreasing for $s \geqslant 64$, it follows that for any $m \geqslant T_0$,

$$\delta^2 \geqslant \frac{64 \log m}{m}.$$

$$\square$$

Now coming back to the proof of Proposition 1, consider an arbitrary $l \in \mathbb{N}$ such that $l > T_0$. Then,

$$\mathbb{P}(\tau \leqslant l) = \mathbb{P}(\tau \leqslant l, \mathcal{Z} > f(T_0)) + \mathbb{P}(\tau \leqslant l, \mathcal{Z} \leqslant f(T_0))$$
$$\leqslant \mathbb{P}(\tau \leqslant l, \mathcal{Z} > f(T_0)) + \Phi(f(T_0)).$$

Note that

$$\mathbb{P}\left(\tau \leqslant l, \mathcal{Z} > f\left(T_0\right)\right) = \mathbb{P}\left(\bigcup_{m=1}^{l}\bigcup_{a,b\in\mathcal{A},a<b}\left\{\mathcal{Z}_{a,b} + \sum_{j=1}^{m}\left(X_{a,j} - X_{b,j}\right) < 4\sqrt{m\log m}, \mathcal{Z}_{a,b} > f\left(T_0\right)\right\}\right)$$

$$\underset{(\dagger)}{=} \mathbb{P}\left(\bigcup_{m=T_0}^{l}\bigcup_{a,b\in\mathcal{A},a<b}\left\{\mathcal{Z}_{a,b} + \sum_{j=1}^{m}\left(X_{a,j} - X_{b,j}\right) < 4\sqrt{m\log m}, \mathcal{Z}_{a,b} > f\left(T_0\right)\right\}\right)$$

$$\leqslant \sum_{m=T_0}^{l}\sum_{a,b\in\mathcal{A},a<b}\mathbb{P}\left(\mathcal{Z}_{a,b} + \sum_{j=1}^{m}\left(X_{a,j} - X_{b,j}\right) < 4\sqrt{m\log m}, \mathcal{Z}_{a,b} > f\left(T_0\right)\right)$$

$$= \sum_{m=T_0}^{l}\sum_{a,b\in\mathcal{A},a<b}\mathbb{P}\left(\mathcal{Z}_{a,b} + \sum_{j=1}^{m}\left(X_{a,j} - X_{b,j} - \delta\right) < -m\left(\delta - 4\sqrt{\frac{\log m}{m}}\right), \mathcal{Z}_{a,b} > f\left(T_0\right)\right)$$

$$\leqslant \sum_{m=T_0}^{l}\sum_{a,b\in\mathcal{A},a<b}\mathbb{P}\left(\sum_{j=1}^{m}\left(X_{a,j} - X_{b,j} - \delta\right) < -m\left(\delta - 4\sqrt{\frac{\log m}{m}}\right), \mathcal{Z}_{a,b} > f\left(T_0\right)\right)$$

$$\underset{(\ddagger)}{\leqslant} \sum_{m=T_0}^{l}\sum_{a,b\in\mathcal{A},a<b}\mathbb{P}\left(\sum_{j=1}^{m}\left(X_{a,j} - X_{b,j} - \delta\right) < -4\sqrt{m\log m}, \mathcal{Z}_{a,b} > f\left(T_0\right)\right)$$

$$= \bar{\Phi}\left(f\left(T_0\right)\right)\sum_{m=T_0}^{l}\sum_{a,b\in\mathcal{A},a<b}\mathbb{P}\left(\sum_{j=1}^{m}\left(X_{a,j} - X_{b,j} - \delta\right) < -4\sqrt{m\log m}\right)$$

$$\underset{(\star)}{\leqslant} \bar{\Phi}\left(f\left(T_0\right)\right)\sum_{m=T_0}^{l}\sum_{a,b\in\mathcal{A},a<b}\frac{1}{m^8}$$

$$\leqslant \bar{\Phi}\left(f\left(T_0\right)\right)K^2\sum_{m=T_0}^{\infty}\frac{1}{m^8}$$

$$\underset{(*)}{\leqslant} \frac{\bar{\Phi}\left(f\left(T_0\right)\right)}{2},$$

where (†) follows from Lemma 1, (‡) from Lemma 2, ($\star$) follows using the Chernoff-Hoeffding bound [15] and finally, ($*$) follows from the definition of $T_0$. Therefore, we have

$$\mathbb{P}\left(\tau \leqslant l\right) \leqslant \frac{\bar{\Phi}\left(f\left(T_0\right)\right)}{2} + \Phi\left(f\left(T_0\right)\right) = 1 - \frac{\bar{\Phi}\left(f\left(T_0\right)\right)}{2}$$

$$\implies \mathbb{P}\left(\tau > l\right) \geqslant \frac{\bar{\Phi}\left(f\left(T_0\right)\right)}{2}.$$

Taking the limit $l \to \infty$ and appealing to the continuity of probability, we obtain

$$\mathbb{P}\left(\tau = \infty\right) \geqslant \frac{\bar{\Phi}\left(f\left(T_0\right)\right)}{2}$$

$$\implies \mathbb{P}\left(\bigcap_{m\geqslant 1}\bigcap_{a,b\in\mathcal{A},a<b}\left\{\left|\mathcal{Z}_{a,b} + \sum_{j=1}^{m}\left(X_{a,j} - X_{b,j}\right)\right| \geqslant 4\sqrt{m\log m}\right\}\right) \geqslant \frac{\bar{\Phi}\left(f\left(T_0\right)\right)}{2}.$$

$$\square$$

## F   Proof of Theorem 3

We will initially assume $\delta > 8\sqrt{\log n/n}$ for technical convenience. In the final step leading up to the asserted bound, we will relax this assumption by offsetting regret appropriately.

Let $\mathcal{A} := \{1, 2, ..., K\}$. Define the following stopping times:

$$\tau_1 := \inf\left\{ m \in \mathbb{N} : \exists a, b \in \mathcal{A}, a < b \text{ s.t. } \left| \mathcal{Z}_{a,b} + \sum_{j=1}^{m} (X_{a,j} - X_{b,j}) \right| < 4\sqrt{m \log m} \right\},$$

$$\tau_2 := \inf\left\{ m \in \mathbb{N} : \left| \sum_{j=1}^{m} (X_{a,j} - X_{b,j}) \right| \geqslant 4\sqrt{m \log n} \ \forall\, a, b \in \mathcal{A}, \ a < b \right\}.$$

Let $R_t$ denote the cumulative pseudo-regret of `CAB-K` (calibrated for a horizon of play $n$) after $t \leqslant n$ pulls. Let $\mathtt{D}$ denote the event that the first batch of $K$ arms queried from the reservoir is "all-distinct," i.e., no two arms in this batch belong to the same type; let $\mathtt{D}^{\mathrm{c}}$ be the complement of this event. Let $\mathtt{CI}$ denote the event that the algorithm commits to an inferior-typed arm. Let $\tilde{R}_t, \bar{R}_t$ be independently drawn from the same distribution as $R_t$. Let $x^+ := \max(x, 0)$ for $x \in \mathbb{R}$. Then, $R_n$ evolves according to the following stochastic recursion:

$R_n$
$$\leqslant \mathbb{1}\{\mathtt{D}\} \left[ \mathbb{1}\{\tau_1 < \tau_2\} \left( \bar{\Delta} \min (K\tau_1, n) + \tilde{R}_{(n-K\tau_1)^+} \right) + \mathbb{1}\{\tau_1 \geqslant \tau_2\} \left( \bar{\Delta} \min (K\tau_2, n) + \mathbb{1}\{\mathtt{CI}\}\bar{\Delta} (n - K\tau_2)^+ \right) \right]$$
$$+ \mathbb{1}\{\mathtt{D}^{\mathrm{c}}\} \left[ \mathbb{1}\{\tau_1 < \tau_2\} \left( \bar{\Delta} \min (K\tau_1, n) + \bar{R}_{(n-K\tau_1)^+} \right) + \mathbb{1}\{\tau_1 \geqslant \tau_2\} \bar{\Delta} n \right]$$
$$\leqslant \mathbb{1}\{\mathtt{D}\} \left[ \mathbb{1}\{\tau_1 < \tau_2\} \left( \bar{\Delta} \min (K\tau_2, n) + \tilde{R}_n \right) + \mathbb{1}\{\tau_1 \geqslant \tau_2\} \left( \bar{\Delta} \min (K\tau_2, n) + \mathbb{1}\{\mathtt{CI}\}\bar{\Delta} n \right) \right]$$
$$+ \mathbb{1}\{\mathtt{D}^{\mathrm{c}}\} \left[ \mathbb{1}\{\tau_1 < \tau_2\} \left( \bar{\Delta} \min (K\tau_1, n) + \bar{R}_n \right) + \mathbb{1}\{\tau_1 \geqslant \tau_2\} \bar{\Delta} n \right]$$
$$\leqslant \mathbb{1}\{\mathtt{D}\} \left[ 2\bar{\Delta} \min (K\tau_2, n) + \mathbb{1}\{\tau_1 < \tau_2\} \tilde{R}_n + \mathbb{1}\{\mathtt{CI}\}\bar{\Delta} n \right] + \mathbb{1}\{\mathtt{D}^{\mathrm{c}}\} \left[ \bar{\Delta} K\tau_1 + \bar{R}_n + \mathbb{1}\{\tau_1 \geqslant \tau_2\} \bar{\Delta} n \right].$$

Taking expectations on both sides, one recovers using the independence of $\tilde{R}_n, \bar{R}_n$ that

$\mathbb{E} R_n$
$$\leqslant \frac{\bar{\Delta}}{\mathbb{P}(\tau_1 \geqslant \tau_2 | \mathtt{D})} \left[ 2\mathbb{E}\left[\min (K\tau_2, n) | \mathtt{D}\right] + \left( \frac{\mathbb{P}(\mathtt{D}^{\mathrm{c}})}{\mathbb{P}(\mathtt{D})} \right) \mathbb{E}\left[K\tau_1 | \mathtt{D}^{\mathrm{c}}\right] + \left( \mathbb{P}(\mathtt{CI}|\mathtt{D}) + \left( \frac{\mathbb{P}(\mathtt{D}^{\mathrm{c}})}{\mathbb{P}(\mathtt{D})} \right) \mathbb{P}(\tau_1 \geqslant \tau_2 | \mathtt{D}^{\mathrm{c}}) \right) n \right],$$
where $\mathbb{P}(\mathtt{D}) = K! \prod_{i=1}^{K} \alpha_i$.

### F.1 Lower bounding $\mathbb{P}(\tau_1 \geqslant \tau_2 | \mathtt{D})$

Note that
$$\mathbb{P}(\tau_1 < \tau_2 | \mathtt{D}) = \mathbb{P}(\tau_1 < \tau_2, \tau_2 = \infty | \mathtt{D}) + \mathbb{P}(\tau_1 < \tau_2, \tau_2 < \infty | \mathtt{D})$$
$$\leqslant \mathbb{P}(\tau_2 = \infty | \mathtt{D}) + \mathbb{P}(\tau_1 < \infty | \mathtt{D})$$
$$= \mathbb{P}(\tau_1 < \infty | \mathtt{D})$$
$$\leqslant 1 - \beta_{\delta, K},$$
where the equality in the third step follows since $\tau_2$ is almost surely finite on the event $\mathtt{D}$, and the final inequality is due to Proposition 1. Thus, $\mathbb{P}(\tau_1 \geqslant \tau_2 | \mathtt{D}) \geqslant \beta_{\delta, K}$.

#### F.1.1 Proof that $\tau_2$ is almost surely finite on $\mathtt{D}$

Let $\mathbb{P}_{\mathtt{D}}(\cdot) := \mathbb{P}(\cdot | \mathtt{D})$ be the conditional measure w.r.t. the event $\mathtt{D}$. Let $\mathcal{A} := \{1, 2, ..., K\}$. Then, by continuity of probability, we have
$$\mathbb{P}_{\mathtt{D}}(\tau_2 = \infty) = \lim_{l \to \infty} \mathbb{P}_{\mathtt{D}}(\tau_2 > l)$$
$$= \lim_{l \to \infty} \mathbb{P}_{\mathtt{D}}\left( \bigcap_{m=1}^{l} \bigcup_{a,b \in \mathcal{A}, a < b} \left\{ \left| \sum_{j=1}^{m} (X_{a,j} - X_{b,j}) \right| < 4\sqrt{m \log n} \right\} \right)$$
$$\leqslant \lim_{l \to \infty} \sum_{a,b \in \mathcal{A}, a < b} \mathbb{P}_{\mathtt{D}}\left( \left| \sum_{j=1}^{l} (X_{a,j} - X_{b,j}) \right| < 4\sqrt{l \log n} \right).$$

On D, it must be that $|\mathbb{E}\left[X_{a,j} - X_{b,j}\right]| \geqslant \delta$. Without loss of generality, assume that $\mathbb{E}\left[X_{a,j} - X_{b,j}\right] \geqslant \delta$. Then,

$$
\begin{aligned}
\mathbb{P}_{\text{D}}\left(\tau_2 = \infty\right) &\leqslant \lim_{l \to \infty} \sum_{a,b \in \mathcal{A}, a < b} \mathbb{P}_{\text{D}}\left(\sum_{j=1}^{l} (X_{a,j} - X_{b,j}) < 4\sqrt{l \log n}\right) \\
&= \lim_{l \to \infty} \sum_{a,b \in \mathcal{A}, a < b} \mathbb{P}_{\text{D}}\left(\sum_{j=1}^{l} (X_{a,j} - X_{b,j} - \delta) < -l\left(\delta - 4\sqrt{\frac{\log n}{l}}\right)\right) \\
&\leqslant \lim_{l \to \infty} \sum_{a,b \in \mathcal{A}, a < b} \mathbb{P}_{\text{D}}\left(\sum_{j=1}^{l} (X_{a,j} - X_{b,j} - \delta) < -4l\sqrt{\log n}\left(\frac{2}{\sqrt{n}} - \frac{1}{\sqrt{l}}\right)\right),
\end{aligned}
$$

where the last inequality follows since $\delta > 8\sqrt{\log n/n}$ (by assumption). Now, using the Chernoff-Hoeffding bound [15] together with the fact that $-1 \leqslant X_{a,j} - X_{b,j} \leqslant 1$, we obtain for $l > n$ and any $a, b \in \mathcal{A}, a < b$ that

$$
\begin{aligned}
\mathbb{P}_{\text{D}}\left(\sum_{j=1}^{l} (X_{a,j} - X_{b,j} - \delta) < -4l\sqrt{\log n}\left(\frac{2}{\sqrt{n}} - \frac{1}{\sqrt{l}}\right)\right) &\leqslant \exp\left[-8l\left(\frac{2}{\sqrt{n}} - \frac{1}{\sqrt{l}}\right)^2 \log n\right] \\
&= \exp\left[-8\left(\frac{4l}{n} - 4\sqrt{\frac{l}{n}} + 1\right)^2 \log n\right].
\end{aligned}
$$

Summing over $a, b \in \mathcal{A}, a < b$ and taking the limit $l \to \infty$ proves the stated assertion. $\qquad \square$

### F.2   Upper bounding $\mathbb{E}\left[\min\left(K\tau_2, n\right)|\text{D}\right]$

Let $\mathbb{P}_{\text{D}}(\cdot) := \mathbb{P}\left(\cdot|\text{D}\right)$ be the conditional measure w.r.t. the event D. Let $\mathcal{A} := \{1, 2, ..., K\}$. Then,

$$
\begin{aligned}
\mathbb{E}\left[\min\left(K\tau_2, n\right)|\text{D}\right] &= K\mathbb{E}\left[\min\left(\tau_2, \frac{n}{K}\right)\Big|\text{D}\right] \\
&\leqslant K\mathbb{E}\left[\min\left(\tau_2, n\right)|\text{D}\right] \\
&\leqslant K + K\sum_{k=2}^{n} \mathbb{P}_{\text{D}}\left(\tau_2 \geqslant k\right) \\
&\leqslant K + K\sum_{k=1}^{n} \mathbb{P}_{\text{D}}\left(\tau_2 \geqslant k+1\right) \\
&\leqslant K + K\sum_{k=1}^{n} \sum_{a,b \in \mathcal{A}, a < b} \mathbb{P}_{\text{D}}\left(\left|\sum_{j=1}^{k} (X_{a,j} - X_{b,j})\right| < 4\sqrt{k \log n}\right).
\end{aligned}
$$

On D, it must be that $|\mathbb{E}\left[X_{a,j} - X_{b,j}\right]| \geqslant \delta$. Without loss of generality, assume that $\mathbb{E}\left[X_{a,j} - X_{b,j}\right] \geqslant \delta$. Then,

$$
\mathbb{E}\left[\min\left(K\tau_2, n\right)| \mathtt{D}\right] \leqslant K + K \sum_{k=1}^{n} \sum_{a,b\in\mathcal{A}, a<b} \mathbb{P}_{\mathtt{D}}\left(\sum_{j=1}^{k}(X_{a,j} - X_{b,j}) < 4\sqrt{k\log n}\right)
$$

$$
= K + K\sum_{k=1}^{n} \sum_{a,b\in\mathcal{A}, a<b} \mathbb{P}_{\mathtt{D}}\left(\sum_{j=1}^{k}(X_{a,j} - X_{b,j} - \delta) < -k\left(\delta - 4\sqrt{\frac{\log n}{k}}\right)\right)
$$

$$
\leqslant K + \frac{32K^3\log n}{\delta^2} + K\sum_{k=\left\lceil\frac{64\log n}{\delta^2}\right\rceil}^{n} \sum_{a,b\in\mathcal{A}, a<b} \mathbb{P}_{\mathtt{D}}\left(\sum_{j=1}^{k}(X_{a,j} - X_{b,j} - \delta) < \frac{-k\delta}{2}\right),
$$

where the last step follows since $\delta > 8\sqrt{\log n/n}$ (by assumption) implies $n > 64\log n/\delta^2$, and $k \geqslant 64\log n/\delta^2$ implies $\delta - 4\sqrt{\log n/k} \geqslant \delta/2$. Finally, using the Chernoff-Hoeffding inequality [15] together with the fact that $|X_{a,j} - X_{b,j}| \leqslant 1$, one obtains

$$
\mathbb{E}\left[\min\left(K\tau_2, n\right)| \mathtt{D}\right] \leqslant K + \frac{32K^3\log n}{\delta^2} + \frac{K^3}{2}\sum_{k=\left\lceil\frac{64\log n}{\delta^2}\right\rceil}^{n}\exp\left(\frac{-\delta^2 k}{8}\right) \leqslant \frac{64K^3\log n}{\delta^2}.
$$

### F.3 Upper bounding $\mathbb{E}\left[K\tau_1| \mathtt{D}^{\mathtt{c}}\right]$

The event $\mathtt{D}^{\mathtt{c}}$ will be implicitly assumed and we will drop the conditional argument for notational simplicity. Let $\mathcal{A} := \{1, 2, ..., K\}$. Without loss of generality, suppose that arm 1 and 2 belong to the same type. Then,

$$
\mathbb{E}\left[K\tau_1| \mathtt{D}^{\mathtt{c}}\right] = K + K\sum_{k\geqslant 2}\mathbb{P}\left(\tau_1 \geqslant k\right)
$$

$$
= K + K\sum_{k\geqslant 1}\mathbb{P}\left(\tau_1 \geqslant k+1\right)
$$

$$
= K + K\sum_{k\geqslant 1}\mathbb{P}\left(\bigcap_{m=1}^{k}\bigcap_{a,b\in\mathcal{A}, a<b}\left\{\left|\mathcal{Z}_{a,b} + \sum_{j=1}^{m}(X_{1,j} - X_{2,j})\right| \geqslant 4\sqrt{m\log m}\right\}\right)
$$

$$
\leqslant K + K\sum_{k\geqslant 1}\mathbb{P}\left(\left|\mathcal{Z}_{1,2} + \sum_{j=1}^{k}(X_{1,j} - X_{2,j})\right| \geqslant 4\sqrt{k\log k}\right).
$$

Since $\mathcal{Z}_{a,b}$ is a standard Gaussian, and the increments $X_{1,j} - X_{2,j}$ are zero-mean sub-Gaussian with variance proxy 1, it follows from the Chernoff-Hoeffding concentration bound [15] that

$$
\mathbb{E}\left[K\tau_1| \mathtt{D}^{\mathtt{c}}\right] \leqslant K + 2K\sum_{k\geqslant 1}\frac{1}{k^4} = \left(1 + \frac{\pi^4}{45}\right)K < 4K.
$$

### F.4 Upper bounding $\mathbb{P}\left(\mathtt{CI}| \mathtt{D}\right)$

Let $\mathbb{P}_{\mathtt{D}}(\cdot) := \mathbb{P}\left(\cdot| \mathtt{D}\right)$ be the conditional measure w.r.t. the event $\mathtt{D}$. Let $\mathcal{A} := \{1, 2, ..., K\}$ and without loss of generality, suppose that arm 1 is optimal (mean $\mu_1$). Then,

$$\mathbb{P}\left(\texttt{CI}\,|\,\texttt{D}\right) \leqslant \mathbb{P}_{\texttt{D}}\left(\bigcup_{b=2}^{K}\left\{\sum_{j=1}^{\tau_2}\left(X_{1,j}-X_{b,j}\right)\leqslant -4\sqrt{\tau_2\log n}\right\}\right)$$

$$\leqslant \sum_{b=2}^{K}\sum_{k=1}^{n}\mathbb{P}_{\texttt{D}}\left(\sum_{j=1}^{k}\left(X_{1,j}-X_{b,j}\right)\leqslant -4\sqrt{k\log n}\right) + \sum_{b=2}^{K}\mathbb{P}_{\texttt{D}}\left(\tau_2 > n\right)$$

$$\leqslant \sum_{b=2}^{K}\sum_{k=1}^{n}\frac{1}{n^8} + \frac{K^3}{n^8}$$

$$\leqslant \frac{K}{n^7} + \frac{K^3}{n^8},$$

where the second-to-last step follows using the Chernoff-Hoeffding inequality [15].

### F.5   Upper bounding $\mathbb{P}\left(\tau_1 \geqslant \tau_2\,|\,\texttt{D}^\texttt{c}\right)$

Let $\mathbb{P}_{\texttt{D}^\texttt{c}}(\cdot) := \mathbb{P}\left(\cdot\,|\texttt{D}^\texttt{c}\right)$ be the conditional measure w.r.t. the event $\texttt{D}^\texttt{c}$. Let $\mathcal{A} := \{1, 2, ..., K\}$. On $\texttt{D}^\texttt{c}$, there exist 2 arms in $\mathcal{A}$ that belong to the same type; without loss of generality suppose that these arms are indexed by $1, 2$. Then,

$$\mathbb{P}\left(\tau_1 \geqslant \tau_2\,|\,\texttt{D}^\texttt{c}\right) \leqslant \mathbb{P}\left(\tau_1 > n\,|\,\texttt{D}^\texttt{c}\right) + \mathbb{P}\left(\tau_2 \leqslant n\,|\,\texttt{D}^\texttt{c}\right)$$

$$\leqslant \frac{2}{n^4} + \mathbb{P}_{\texttt{D}^\texttt{c}}\left(\tau_2 \leqslant n\,|\,\texttt{D}^\texttt{c}\right)$$

$$\leqslant \frac{2}{n^4} + \mathbb{P}_{\texttt{D}^\texttt{c}}\left(\bigcup_{m=1}^{n}\bigcap_{a,b\in\mathcal{A},a<b}\left\{\left|\sum_{j=1}^{m}\left(X_{a,j}-X_{b,j}\right)\right|\geqslant 4\sqrt{m\log n}\right\}\right)$$

$$\leqslant \frac{2}{n^4} + \sum_{m=1}^{n}\mathbb{P}_{\texttt{D}^\texttt{c}}\left(\left|\sum_{j=1}^{m}\left(X_{1,j}-X_{2,j}\right)\right|\geqslant 4\sqrt{m\log n}\right)$$

$$\leqslant \frac{2}{n^4} + \frac{2}{n^7}, \tag{13}$$

where the last step follows using the Chernoff-Hoeffding bound [15].

### F.6   Putting everything together

Combining everything, one finally obtains that when $\delta > 8\sqrt{\log n/n}$,

$$\mathbb{E}R_n \leqslant \frac{CK^3\bar{\Delta}}{\beta_{\delta,K}}\left(\frac{\log n}{\delta^2} + \frac{1}{\mathbb{P}\left(\texttt{D}\right)}\right),$$

where $\beta_{\delta,K}$ is as defined in (2), $\mathbb{P}\left(\texttt{D}\right) = K!\prod_{i=1}^{K}\alpha_i$, and $C$ is some absolute constant. When $\delta \leqslant 8\sqrt{\log n/n}$, regret is at most $\bar{\Delta}n \leqslant 64\bar{\Delta}/\delta^2\log n$. Thus, the aforementioned bound, in fact, holds generally for some large enough absolute constant $C$. $\qquad\square$

## G   Exponential doubling for "anytime" algorithms with logarithmic regret

Below, we propose an exponential doubling trick that can be used to make algorithms horizon-free while preserving their instance-dependent regret upper bounds in the leading order. The reader is referred to [5] for a comprehensive survey of standard doubling tricks used in the literature.

---

**Algorithm 3** HF($\mathfrak{A}$) (EXPONENTIAL DOUBLING)

---

1: **Input:** Algorithm $\mathfrak{A}$.
2: **Generate doubling sequence:** $T_{-1} = 0$; $T_i = 2^{2^i}$ for $i = 0, 1, ....$
3: **for** $i \in \{0, 1, ...\}$ **do**
4:     Specify as input to $\mathfrak{A}$ a horizon of play of $T_i - T_{i-1}$.
5:     Restart $\mathfrak{A}$ at time $t = T_{i-1} + 1$ and run until $t = T_i$.

---

**Theorem 7 (Preservation of logarithmic regret under HF($\mathfrak{A}$))** *Consider an algorithm $\mathfrak{A}$ (possibly horizon-dependent) with an expected regret of $\mathbb{E}R_n^{\mathfrak{A}} \leqslant C \log n + D$ at the end of a horizon of $n \geqslant 1$ plays, where $C$ and $D$ are non-negative constants. Then, the expected regret of the horizon-free policy $\pi$ given by HF($\mathfrak{A}$) after any number $n \geqslant 1$ of plays is bounded as*

$$\mathbb{E}R_n^{\pi} \leqslant 80 \left[ C \log n + D \log \log(n + 2) \right].$$

**Discussion.** The proof is provided in Appendix G.1. It is possible to improve the multiplicative constants in Theorem 7 via a finer calibration of the doubling sequence $(T_i : i = 0, 1, ...)$. However, our goal simply is to design horizon-free policies for the matching problem that are rate-optimal in $n$ modulo constant multiplicative factors and to that end, Theorem 7 serves its purpose. We now state an anytime upper bound on the regret of the horizon-free version of CAB-K.

**Theorem 8 (Upper bound on the regret of HF(CAB-K))** *The expected regret of the policy $\pi$ given by HF(CAB-K) after any number $n \geqslant 1$ of plays is bounded as*

$$\mathbb{E}R_n^{\pi} \leqslant \frac{CK^3\bar{\Delta}}{\beta_{\delta,K}} \left( \frac{\log n}{\delta^2} + \frac{\log \log (n + 2)}{K! \prod_{i=1}^{K} \alpha_i} \right),$$

*where $\beta_{\delta,K}$ is as defined in (2) and $C$ is some absolute constant.*

## G.1   Proof of Theorem 7

In what follows, $\log$ corresponds to the natural logarithm unless a base is explicitly specified otherwise. Define $\Lambda_n := \min\{i \in \mathbb{N} : T_i \geqslant n\} = \lceil \log_2 \log_2 n \rceil$. Since the algorithm $\mathfrak{A}$ is restarted at times $\{T_i + 1 : i = -1, 0, 1, ...\}$, where $T_{-1} := 0$, it follows that the expected cumulative regret of the horizon-free policy $\pi$ given by HF($\mathfrak{A}$) after $n$ plays is bounded as

$$
\begin{aligned}
\mathbb{E}R_n^{\pi} \leqslant \sum_{i=0}^{\Lambda_n} \mathbb{E}R_{T_i - T_{i-1}}^{\mathfrak{A}} \leqslant \sum_{i=0}^{\Lambda_n} \left( C \log T_i + D \right) &= \sum_{i=0}^{\Lambda_n} \left( \frac{C}{\log_2 e} \log_2 T_i + D \right) \\
&\leqslant \sum_{i=0}^{\Lambda_n} \left( C2^i + D \right) \\
&= C2^{\Lambda_n + 1} + D \left( \Lambda_n + 1 \right) - C \\
&= 4C \log_2 n + D \log_2 \log_2 n + 2D - C \\
&\leqslant 8C \log n + 2D \log \log n + 3D - C \\
&\leqslant 8C \log n + 80D \log \log(n + 2).
\end{aligned}
$$

$\square$

## G.2   Proof of Theorem 8

This proof is a direct application of Theorem 7. $\square$

# H   Proof of Theorem 4

Note that there are at most $M |\mathcal{J}|$ active threads of HF(CAB-K) at any time. Since HF(CAB-K) is horizon-free, it naturally follows that the regret incurred under $\pi$ is dominated by that in the scenario where all $M |\mathcal{J}|$ threads are active at each $t \in \{1, ..., n\}$. The assertion is now immediate. $\square$

# I  A first-order optimal algorithm for countable-armed bandits with $K \geqslant 2$

---

**Algorithm 4** `CAB-K(UCB)`: Nested UCB1 for K types

---

1: **Initialize new epoch** (resets clock $t \leftarrow 0$): Query $K$ *new* arms; call it set $\mathcal{A} = \{1, 2, ..., K\}$.
2: Play each arm in $\mathcal{A}$ once; observe rewards $\{X_{a,1} : a \in \mathcal{A}\}$.
3: Minimum per-arm sample count $m \leftarrow 1$.
4: Generate $\binom{K}{2}$ independent standard Gaussian random variables $\{\mathcal{Z}_{a,b} : a, b \in \mathcal{A}, a < b\}$.
5: **for** $t \in \{K + 1, K + 2, ...\}$ **do**
6:     **if** $\exists\, a, b \in \mathcal{A}, a < b$ s.t. $\left| \mathcal{Z}_{a,b} + \sum_{j=1}^{m} (X_{a,j} - X_{b,j}) \right| < 4\sqrt{m \log m}$ **then**
7:         Permanently discard $\mathcal{A}$ and repeat from step (1).
8:     **else**
9:         Play arm $a_t \in \arg\max_{a \in \mathcal{A}} \left( \frac{\sum_{j=1}^{N_a(t-1)} X_{a,j}}{N_a(t-1)} + \sqrt{\frac{2 \log(t-1)}{N_a(t-1)}} \right)$.
10:         Observe reward $X_{a_t, N_{a_t}(t)}$.
11:         **if** $m < \min_{a \in \mathcal{A}} N_a(t)$ **then**
12:             $m \leftarrow m + 1$.

---

**Theorem 9 (Upper bound on the regret of CAB-K(UCB))** *The expected regret of the policy $\pi$ given by* `CAB-K(UCB)` *after any number $n \geqslant 1$ of plays is bounded as*

$$\mathbb{E}R_n^\pi \leqslant \frac{CK}{\beta_{\delta,K}} \left( \frac{\log n}{\underline{\Delta}} + \bar{\Delta} \right) + o\left( \frac{\bar{\Delta} n}{\beta_{\delta,K} \prod_{i=1}^{K} \alpha_i} \right),$$

*where $C$ is some absolute constant, $\beta_{\delta,K}$ is as defined in (2), and the little-Oh is asymptotic in $n$ and only hides multiplicative factors in $K$.*

The proof is provided in §I.1.

**Remark 1** *The upper bound is logarithmic in $n$ for $K = 2$; refer to Theorem 5. Whether this would hold also for general $K$ remains an open problem.*

## I.1  Proof of Theorem 9

Let $\mathcal{A} := \{1, 2, ..., K\}$ be the collection of $K$ arms queried during the first epoch. Consider an arbitrary $l \in \mathbb{N}$ s.t. $l \geqslant K$ and define the following:

$$M_l := \min_{a \in \mathcal{A}} N_a(l),$$

$$\tau := \inf \left\{ l \geqslant K : \exists\, a, b \in \mathcal{A}, a < b \text{ s.t. } \left| \mathcal{Z}_{a,b} + \sum_{j=1}^{M_l} (X_{a,j} - X_{b,j}) \right| < 4\sqrt{M_l \log M_l} \right\}, \quad (14)$$

where $N_a(l)$ denotes the sample count from arm $a$ under `UCB1` until time $l$. Note that $\tau$ marks the termination of epoch 1.

Let $R_n$ denote the cumulative pseudo-regret of `CAB-K(UCB)` after $n$ pulls (superscript $\pi$ suppressed for notational convenience). Let $S_n$ denote the cumulative pseudo-regret of `UCB1` after $n$ pulls in a $K$-armed bandit with means $\mu_1 > \mu_2 > ... > \mu_K$. Let D denote the event that the $K$ arms queried in epoch 1 have distinct types (no two belong to the same type). Similarly, let OPT denote the event that the $K$ arms have optimal types (type 1). Let $\tilde{R}_n, \hat{R}_n, \bar{R}_n$ be independently drawn from the same distribution as $R_n$. Then, the evolution of $R_n$ satisfies

$$R_n \leqslant \mathbb{1}\{\text{D}\} \left[ S_{\min(\tau,n)} + \tilde{R}_{(n-\tau)^+} \right] + \mathbb{1}\{\text{D}^c \backslash \text{OPT}\} \left[ \bar{\Delta} \min(\tau, n) + \hat{R}_{(n-\tau)^+} \right] + \mathbb{1}\{\text{OPT}\} \bar{R}_{(n-\tau)^+}$$

$$\underset{(\dagger)}{\leqslant} \mathbb{1}\{\text{D}\} \left[ S_n + \mathbb{1}\{\tau < n\} \tilde{R}_n \right] + \mathbb{1}\{\text{D}^c \backslash \text{OPT}\} \bar{\Delta} \min(\tau, n) + \mathbb{1}\{\text{D}^c \backslash \text{OPT}\} \hat{R}_n + \mathbb{1}\{\text{OPT}\} \bar{R}_n$$

$$\leqslant \mathbb{1}\{\text{D}\} \left[ S_n + \mathbb{1}\{\tau < \infty\} \tilde{R}_n \right] + \mathbb{1}\{\text{D}^c \backslash \text{OPT}\} \bar{\Delta} \min(\tau, n) + \mathbb{1}\{\text{D}^c \backslash \text{OPT}\} \hat{R}_n + \mathbb{1}\{\text{OPT}\} \bar{R}_n,$$

where (†) follows since `CAB-K(UCB)` is agnostic to $n$, and hence the pseudo-regret $R_n$ is weakly increasing in $n$. Taking expectations on both sides, one recovers using the independence of $\tilde{R}_n, \hat{R}_n, \bar{R}_n$ that

$$\mathbb{E}R_n \leqslant \frac{1}{\mathbb{P}\left(\tau = \infty \mid \mathtt{D}\right)}\left[\mathbb{E}S_n + \left(\frac{\bar{\Delta}\mathbb{P}\left(\mathtt{D}^\mathtt{c}\backslash\mathtt{OPT}\right)\mathbb{E}\left[\min\left(\tau, n\right) \mid \mathtt{D}^\mathtt{c}\backslash\mathtt{OPT}\right]}{\mathbb{P}(\mathtt{D})}\right)\right]$$

$$\leqslant \frac{1}{\beta_{\delta,K}}\left[\mathbb{E}S_n + \left(\frac{\bar{\Delta}\mathbb{P}\left(\mathtt{D}^\mathtt{c}\backslash\mathtt{OPT}\right)\mathbb{E}\left[\min\left(\tau, n\right) \mid \mathtt{D}^\mathtt{c}\backslash\mathtt{OPT}\right]}{\mathbb{P}(\mathtt{D})}\right)\right], \tag{15}$$

where $\mathbb{P}(\mathtt{D}) = K!\prod_{i=1}^{K}\alpha_i$, and the last inequality follows using Lemma 3 with $\beta_{\delta,K}$ as defined in (2). We know that $\mathbb{E}S_n \leqslant CK\left(\log n/\underline{\Delta} + \bar{\Delta}\right)$ for some absolute constant $C$ [1]. The rest of the proof is geared towards showing that $\mathbb{E}\left[\min\left(\tau, n\right) \mid \mathtt{D}^\mathtt{c}\backslash\mathtt{OPT}\right] = o(n)$.

### I.1.1  Proof of $\mathbb{E}\left[\min\left(\tau, n\right) \mid \mathtt{D}^\mathtt{c}\backslash\mathtt{OPT}\right] = o(n)$

Let $\mathbb{P}_{\mathtt{D}^\mathtt{c}\backslash\mathtt{OPT}}(\cdot) := \mathbb{P}(\cdot \mid \mathtt{D}^\mathtt{c}\backslash\mathtt{OPT})$ be the conditional measure w.r.t. the event $\mathtt{D}^\mathtt{c}\backslash\mathtt{OPT}$. On $\mathtt{D}^\mathtt{c}\backslash\mathtt{OPT}$, there exist two arms in the consideration set $\mathcal{A}$ that belong to the same type. Without loss of generality, suppose that these are indexed by $1, 2$. Then,

$$\mathbb{E}\left[\min\left(\tau, n\right) \mid \mathtt{D}^\mathtt{c}\backslash\mathtt{OPT}\right] \leqslant K + \sum_{k=K+1}^{n}\mathbb{P}_{\mathtt{D}^\mathtt{c}\backslash\mathtt{OPT}}\left(\tau \geqslant k\right)$$

$$\leqslant K + \sum_{k=K}^{n}\mathbb{P}_{\mathtt{D}^\mathtt{c}\backslash\mathtt{OPT}}\left(\tau \geqslant k+1\right)$$

$$= K + \sum_{k=K}^{n}\mathbb{P}_{\mathtt{D}^\mathtt{c}\backslash\mathtt{OPT}}\left(\bigcap_{l=1}^{k}\bigcap_{a,b\in\mathcal{A},a<b}\left\{\left|\mathcal{Z}_{a,b} + \sum_{j=1}^{M_l}\left(X_{a,j} - X_{b,j}\right)\right| \geqslant 4\sqrt{M_l\log M_l}\right\}\right)$$

$$\leqslant K + \sum_{k=K}^{n}\mathbb{P}_{\mathtt{D}^\mathtt{c}\backslash\mathtt{OPT}}\left(\left|\mathcal{Z}_{1,2} + \sum_{j=1}^{M_k}\left(X_{1,j} - X_{2,j}\right)\right| \geqslant 4\sqrt{M_k\log M_k}\right)$$

$$= K + \sum_{k=K}^{n}\sum_{m=1}^{k}\mathbb{P}_{\mathtt{D}^\mathtt{c}\backslash\mathtt{OPT}}\left(\left|\mathcal{Z}_{1,2} + \sum_{j=1}^{M_k}\left(X_{1,j} - X_{2,j}\right)\right| \geqslant 4\sqrt{M_k\log M_k}, M_k = m\right)$$

$$\underset{(\dagger)}{=} K + \sum_{k=K}^{n}\sum_{m=f(k)}^{k}\mathbb{P}_{\mathtt{D}^\mathtt{c}\backslash\mathtt{OPT}}\left(\left|\mathcal{Z}_{1,2} + \sum_{j=1}^{M_k}\left(X_{1,j} - X_{2,j}\right)\right| \geqslant 4\sqrt{M_k\log M_k}, M_k = m\right)$$

$$\leqslant K + \sum_{k=K}^{n}\sum_{m=f(k)}^{k}\mathbb{P}_{\mathtt{D}^\mathtt{c}\backslash\mathtt{OPT}}\left(\left|\mathcal{Z}_{1,2} + \sum_{j=1}^{m}\left(X_{1,j} - X_{2,j}\right)\right| \geqslant 4\sqrt{m\log m}\right)$$

$$\underset{(\ddagger)}{\leqslant} K + 2\sum_{k=K}^{n}\sum_{m=f(k)}^{k}\frac{1}{m^4}$$

$$= K + 2\sum_{k=K}^{n}\left(\frac{1}{\left(f(k)\right)^4} + \sum_{m=f(k)+1}^{k}\frac{1}{m^4}\right)$$

$$\leqslant K + 2\sum_{k=K}^{n}\left(\frac{1}{\left(f(k)\right)^4} + \frac{1}{3\left(f(k)\right)^3}\right),$$

where (†) follows from Lemma 4, and (‡) using the Chernoff-Hoeffding bound [15]. Since $f(k)$ is monotone non-decreasing and coercive in $k$, it follows that $\mathbb{E}\left[\min\left(\tau, n\right) \mid \mathtt{D}^\mathtt{c}\backslash\mathtt{OPT}\right] = o(n)$, where the little-Oh only hides dependence on $K$. $\qquad\square$

## I.2 Proof of Theorem 5

We know from (15) that

$$\mathbb{E}R_n \leqslant \frac{1}{\beta_{\underline{\Delta},2}} \left[ \mathbb{E}S_n + \left( \frac{\underline{\Delta}\mathbb{P}\left(\texttt{D}^c\backslash\texttt{OPT}\right)\mathbb{E}\left[\min\left(\tau,n\right)|\,\texttt{D}^c\backslash\texttt{OPT}\right]}{\mathbb{P}(\texttt{D})} \right) \right],$$

where $\tau$ is as defined in (14), $\mathbb{P}(\texttt{D}) = 2\alpha_1(1-\alpha_1)$, $\mathbb{P}\left(\texttt{D}^c\backslash\texttt{OPT}\right) = (1-\alpha_1)^2$, and $\mathbb{E}S_n \leqslant C\left(\log n/\underline{\Delta} + \underline{\Delta}\right)$ for some absolute constant $C$ [1]. The rest of the proof is geared towards showing that $\mathbb{E}\left[\min\left(\tau,n\right)|\,\texttt{D}^c\backslash\texttt{OPT}\right] \leqslant \mathbb{E}\left[\tau|\,\texttt{D}^c\backslash\texttt{OPT}\right] \leqslant C'$ (for some absolute constant $C'$). Going forward, we will assume that the probabilities are implicitly conditional to avoid overloading notation. Then, note that

$$\mathbb{E}\left[\tau|\,\texttt{D}^c\backslash\texttt{OPT}\right] = 1 + \sum_{k\geqslant 2}\mathbb{P}\left(\tau\geqslant k\right)$$

$$= 1 + \sum_{k\geqslant 2}\mathbb{P}\left(\bigcap_{l=1}^{k-1}\left\{\left|\mathcal{Z}_{1,2} + \sum_{j=1}^{M_l}\left(X_{1,j} - X_{2,j}\right)\right| \geqslant 4\sqrt{M_l\log M_l}\right\}\right)$$

$$\leqslant 1 + \sum_{k\geqslant 1}\mathbb{P}\left(\left|\mathcal{Z}_{1,2} + \sum_{j=1}^{M_k}\left(X_{1,j} - X_{2,j}\right)\right| \geqslant 4\sqrt{M_k\log M_k}\right)$$

$$= 1 + \sum_{k\geqslant 1}\sum_{m=1}^{k}\mathbb{P}\left(\left|\mathcal{Z}_{1,2} + \sum_{j=1}^{M_k}\left(X_{1,j} - X_{2,j}\right)\right| \geqslant 4\sqrt{M_k\log M_k}, N_1(k) = m\right)$$

$$= 1 + \sum_{k\geqslant 1}\sum_{1\leqslant m\leqslant k/2}\mathbb{P}\left(\left|\mathcal{Z}_{1,2} + \sum_{j=1}^{m}\left(X_{1,j} - X_{2,j}\right)\right| \geqslant 4\sqrt{m\log m}, N_1(k) = m\right)$$

$$+ \sum_{k\geqslant 1}\sum_{k/2<m\leqslant k}\mathbb{P}\left(\left|\mathcal{Z}_{1,2} + \sum_{j=1}^{(k-m)}\left(X_{1,j} - X_{2,j}\right)\right| \geqslant 4\sqrt{(k-m)\log(k-m)}, N_1(k) = m\right)$$

$$= 1 + \sum_{k\geqslant 1}\sum_{1\leqslant m\leqslant k/2}\mathbb{P}\left(\left|\mathcal{Z}_{1,2} + \sum_{j=1}^{m}\left(X_{1,j} - X_{2,j}\right)\right| \geqslant 4\sqrt{m\log m}, N_1(k) = m\right)$$

$$+ \sum_{k\geqslant 1}\sum_{1\leqslant m<k/2}\mathbb{P}\left(\left|\mathcal{Z}_{1,2} + \sum_{j=1}^{m}\left(X_{1,j} - X_{2,j}\right)\right| \geqslant 4\sqrt{m\log m}, N_2(k) = m\right)$$

$$\leqslant 1 + 2\sum_{k\geqslant 1}\sum_{\theta k\leqslant m\leqslant k/2}\mathbb{P}\left(\left|\mathcal{Z}_{1,2} + \sum_{j=1}^{m}\left(X_{1,j} - X_{2,j}\right)\right| \geqslant 4\sqrt{m\log m}\right)$$

$$+ \sum_{k\geqslant 1}\left[\mathbb{P}\left(N_1(k)\leqslant\theta k\right) + \mathbb{P}\left(N_2(k)\leqslant\theta k\right)\right],$$

where $\theta = 1/2 - \sqrt{15}/8$. Using Theorem 4(i) of [18] with $\epsilon = \sqrt{15}/8$, one obtains

$$\mathbb{E}\left[\tau|\,\texttt{D}^c\backslash\texttt{OPT}\right] \leqslant 1 + 2\sum_{k\geqslant 1}\sum_{\theta k\leqslant m\leqslant k/2}\mathbb{P}\left(\left|\mathcal{Z}_{1,2} + \sum_{j=1}^{m}\left(X_{1,j} - X_{2,j}\right)\right| \geqslant 4\sqrt{m\log m}\right) + 16\sum_{k\geqslant 1}\frac{1}{k^2}$$

$$\leqslant 1 + 4\sum_{k\geqslant 1}\sum_{\theta k\leqslant m\leqslant k/2}\frac{1}{m^4} + 16\sum_{k\geqslant 1}\frac{1}{k^2}$$

$$\leqslant 1 + \frac{4}{\theta^4}\sum_{k\geqslant 1}\frac{1}{k^3} + 16\sum_{k\geqslant 1}\frac{1}{k^2}$$

$$=: C'.$$

$$\square$$

# J  Auxiliary results used in the analysis of CAB-K(UCB)

**Lemma 3 (Persistence of heterogeneous consideration sets)** *Consider a $K$-armed bandit with rewards bounded in $[0,1]$ and means $\mu_1 > \mu_2 > ... > \mu_K$. Let $\{X_{a,j} : j = 1, 2, ...\}$ denote the rewards collected from arm $a \in \{1, ..., K\} =: \mathcal{A}$ by* `UCB1` *[1]. Let $\{\mathcal{Z}_{a,b} : a, b \in \mathcal{A}, a < b\}$ be a collection of $\binom{K}{2}$ independent standard Gaussian random variables. Let $N_a(n)$ be the sample count of arm $a$ under* `UCB1` *until time $n$. Define*

$$M_l := \min_{a \in \mathcal{A}} N_a(l),$$

$$\tau := \inf \left\{ l \geqslant K : \exists\, a, b \in \mathcal{A}, a < b \text{ s.t. } \left| \mathcal{Z}_{a,b} + \sum_{j=1}^{M_l} (X_{a,j} - X_{b,j}) \right| < 4\sqrt{M_l \log M_l} \right\}.$$

*Then, $\mathbb{P}\left(\tau = \infty\right) > \beta_{\delta, K}$, where $\beta_{\delta, K}$ is as defined in (2).*

**Lemma 4 (Path-wise lower bound on the arm-sampling rate of UCB1)** *Consider a $K$-armed bandit with rewards bounded in $[0,1]$. Let $N_a(n)$ be the sample count of arm $a \in \{1, ..., K\} =: \mathcal{A}$ under* `UCB1` *[1] until time $n$. Then, for all $n \geqslant K$,*

$$M_n := \min_{a \in \mathcal{A}} N_a(n) \geqslant f(n),$$

*where $(f(n) : n = K, K+1, ...)$ is some deterministic monotone non-decreasing integer-valued sequence satisfying $f(K) = 1$ and $f(n) \to \infty$ as $n \to \infty$.*

## J.1  Proof of Lemma 3

Suppose that there exists a sample-path on which some non-empty subset of arms $\mathfrak{A} \subset \mathcal{A}$ receives a bounded number of pulls asymptotically in the horizon of play. Also suppose that $\mathfrak{A}$ is the maximal such subset, i.e., each arm in $\mathcal{A} \backslash \mathfrak{A}$ is played infinitely often asymptotically on said sample-path. This implies that the UCB score of any arm in $\mathcal{A} \backslash \mathfrak{A}$ is of the order $o\left(\sqrt{\log t}\right)$ at time $t$ (the empirical mean term remains bounded in $[0,1]$ and therefore can be ignored). At the same time, the boundedness hypothesis implies that the UCB score of any arm in $\mathfrak{A}$ will grow as $\Omega\left(\sqrt{\log t}\right)$. Thus, for $t$ large enough, UCB scores of arms in $\mathfrak{A}$ will start to dominate those in $\mathcal{A} \backslash \mathfrak{A}$ and the algorithm will end up playing an arm from $\mathfrak{A}$ at some point, thus increasing the cumulative sample-count of arms in $\mathfrak{A}$ by 1. As $t$ grows further, one can replicate the preceding argument an arbitrary number of times to conclude that $\mathfrak{A}$ receives an unbounded number of pulls on the sample-path under consideration, thereby contradicting the boundedness hypothesis. Therefore, it must be the case that each arm in $\mathcal{A}$ is played infinitely often on *every* sample-path. Consequently, $M_n = \min_{a \in \mathcal{A}} N_a(n) \to \infty$ as $n \to \infty$ on every sample-path.

Now since $(M_n : n = K, K+1, ...)$ is an integer-valued process (starting from $M_K = 1$) with unit increments (wherever they exist), it follows that on every sample-path, $\tau$, in fact, weakly dominates the stopping time $\tau'$ given by

$$\tau' := \inf \left\{ m \in \mathbb{N} : \exists\, a, b \in \mathcal{A}, a < b \text{ s.t. } \left| \mathcal{Z}_{a,b} + \sum_{j=1}^{m} (X_{a,j} - X_{b,j}) \right| < 4\sqrt{m \log m} \right\}. \quad (16)$$

Therefore, $\mathbb{P}\left(\tau = \infty\right) \geqslant \mathbb{P}\left(\tau' = \infty\right) > \beta_{\delta, K}$; the last inequality follows from Proposition 1.  $\square$

## J.2  Proof of Lemma 4

Suppose that $\mathcal{S}_n = \{(N_a(n) : a \in \mathcal{A})\}$ denotes the set of possible sample-count realizations under `UCB1` when the horizon of play is $n$. Define $f(n) := \min_{(N_a(n):a \in \mathcal{A}) \in \mathcal{S}_n} \min_{a \in \mathcal{A}} N_a(n)$. Since $\mathcal{S}_n$ is finite, aforementioned minimum is attained at some $(N_a^*(n) : a \in \mathcal{A}) \in \mathcal{S}_n$. Note that $(N_a^*(n) : a \in \mathcal{A})$ is not a random vector as it corresponds to a specific set of sample-paths (possibly non-unique) on which $\min_{a \in \mathcal{A}} N_a(n)$ is minimized. Therefore, $f(n) = \min_{a \in \mathcal{A}} N_a^*(n)$ is deterministic. We have already established in the proof of Lemma 3 that for each $a \in \mathcal{A}$, $N_a(n) \to \infty$ as $n \to \infty$ on every sample-path. In particular, this also implies $N_a^*(n) \to \infty$ as $n \to \infty$. Thus, we have established the existence of a sequence $f(n)$ satisfying the assertions of the lemma.  $\square$

# K    More Explore-then-Commit policies for countable-armed bandits

---

**Algorithm 5** ETC-$\infty$(K)

---

1: **Input:** (i) Horizon of play $n \geqslant K$, (ii) A lower bound $\underline{\delta} \in (0, \delta]$ on the minimal reward gap $\delta$.
2: Set budget $T = n$.
3: **Initialize new epoch:** Query $K$ *new* arms; call it consideration set $\mathcal{A} = \{1, 2, ..., K\}$.
4: Set exploration duration $L = \left\lceil 2\underline{\delta}^{-2} \log n \right\rceil$.
5: $m \leftarrow \min\left(L, \lfloor T/K \rfloor\right)$.
6: Play each arm in $\mathcal{A}$ $m$ times; observe rewards $\{(X_{1,j}, ..., X_{K,j}) : j = 1, ..., m\}$.
7: Update budget: $T \leftarrow T - Km$.
8: **if** $\exists\, a, b \in \mathcal{A}$, $a < b$ s.t. $\left| \sum_{j=1}^{m}(X_{a,j} - X_{b,j}) \right| < \underline{\delta}m$ **then**
9:     Permanently discard $\mathcal{A}$, and repeat from step (3).
10: **else**
11:     Permanently commit to arm $a^* \in \arg\max_{a \in \mathcal{A}} \left\{ \sum_{j=1}^{m} X_{a,j} \right\}$.

---

---

**Algorithm 6** ETC-RAW

---

1: **Input:** Horizon of play $n \geqslant K$.
2: Set budget $T = n$; set epoch counter $k = 1$.
3: **Initialize new epoch:** Query $K$ *new* arms; call it consideration set $\mathcal{A} = \{1, 2, ..., K\}$.
4: Set exploration duration $L = \left\lceil e^{2\sqrt{k}} \log n \right\rceil$.
5: $m \leftarrow \min\left(L, \lfloor T/K \rfloor\right)$.
6: Play each arm in $\mathcal{A}$ $m$ times; observe rewards $\{(X_{1,j}, ..., X_{K,j}) : j = 1, ..., m\}$.
7: Update budget: $T \leftarrow T - Km$.
8: **if** $\exists\, a, b \in \mathcal{A}$, $a < b$ s.t. $\left| \sum_{j=1}^{m}(X_{a,j} - X_{b,j}) \right| < 2me^{-\sqrt{k}}$ **then**
9:     Permanently discard $\mathcal{A}$.
10:     $k \leftarrow k + 1$.
11:     Repeat from step (3).
12: **else**
13:     Permanently commit to arm $a^* \in \arg\max_{a \in \mathcal{A}} \left\{ \sum_{j=1}^{m} X_{a,j} \right\}$.

---