# OpenReview forum: "Dynamic Learning in Large Matching Markets"
_NeurIPS.cc/2022/Conference — NeurIPS 2022 Accept_

### Official Review · Reviewer_ZuWy · 2022-07-03

**Rating:** 5
**Confidence:** 3
**Soundness:** 3 good
**Presentation:** 2 fair
**Contribution:** 3 good

**Summary:**

This paper studies a sequential matching problem where there are unlimited supply of workers whose attributes are drawn from a population distribution. This is different from existing matching bandit literature which considered finite markers and hence focused on competition and congestion in dynamic matching. On the other hand, this paper considers choice overload issue in large markets. The authors established lower bounds on the regret of any matching algorithm in this setting and proposed a rate-optimal learning algorithm.


**Questions:**

See the comments in the Weakness part.

**Ethics Review Area:**

["I don’t know"]

**Limitations:**

There is no discussion on the potential negative societal impact of their work.

**Strengths And Weaknesses:**

Strengths

(1) Dynamic matching under uncertainty is an interesting and new research direction.

(2) The proved regret lower bound is useful to understand the limit of the considered matching problem.


Weaknesses

(1) Motivation is not well justified. This paper considers the sequential matching problem with unlimited supply of workers. The unlimited nature is a key difference to existing dynamic matching algorithms with finite markets. However, it is not fully convinced that the considered matching problem is realistic in practice. In real matching problems, we typically have finite markets. It would be important to add a convincing real example to motivate the considered setting. In addition, given such real example, it is also important to discuss why existing dynamic matching methods with finite markets could not handle it.


(2) Writing. The current version is not easy to follow due to the theoretical nature of this paper. In addition, there are many confusing parts in the theoretical analysis and the algorithms.

First, Section 3 consist of two different lower bounds for the considered problem and both theorems have conflicting components. In the lower bound of Theorem 1, the regret lower bound is a reverse function of the mean reward gap $\Delta_j$. However, in the lower bound of Theorem 2, the regret lower bound is a linear function to $\Delta_j$. These two lower bounds are proved under different assumptions. A meaningful lower bound is to consider the same problem setting used in the upper bound in order to justify the rate-optimality of the proposed algorithm.

Second, the authors proposed Algorithm 2 (CAB-K) in the main paper and proved its nice $log(n)$ regret upper bound. But in the experiments, the proposed CAB-K was not superior over existing algorithms (e.g., UCB1 and Sampling-UCB as shown in Figure 2). So they included another new algorithm (CAB-K (UCB) in Algorithm 4) which indeed performed very well in practice. However, the authors only proved a $o(n)$ regret upper bound for CAB-K (UCB), and it is unclear if the log(n) regret can be achieved for general $K$. The algorithms are confusing to follow due to mixed algorithms with mixed theoretical properties.

Third, because of the assumption on unlimited supply of workers, the dynamic matching problem eventually reduces to a countably many-armed bandit problem. This is different from existing dynamic matching literature which considered finite markers and hence focused on competition and congestion in dynamic matching. Therefore, one suggestion is to position this paper as a contribution to the countably many-armed bandit literature by proposing new CAB-K or CAK-K (UCB) algorithms.

In summary, because of the above reasons and the 9-page limit of Neurips, a journal might be a better fit to this paper.

---

> ### Author Response · Authors · 2022-08-02
> **Response to reviewer ZuWy**
>
> Thank you for your observations and feedback! We will attempt to systematically address your concerns below.
>
> (I) MOTIVATION. While the online matching problem has long existed in theoretical computer science where the objective is to achieve optimal or near-optimal "competitive ratios," only a handful of recent papers (see, e.g., reference [22] in the paper and subsequent follow up works) study the problem under a statistical learning framework. In the aforementioned stream of literature, the focus, by and large, is on learning a "stable" matching in decentralized markets where agents submit noisy preferences to the platform at each round (potentially strategically) while the platform simply deploys a stable matching algorithm like "Deferred Acceptance" to assign matches.
>
> In contrast, our work studies large centralized markets where the platform's operational objective is to learn a "maximal" matching (as opposed to stable), and understand how "choice overload" affects achievable performance. To our best knowledge, our work is the first to study such settings, and a large market regime is a fitting lens for a principled analysis of these aspects. Furthermore, an unlimited supply of agents endowed with a population-level distribution is a fairly standard modeling choice also in the operations research literature, see, e.g., reference [17] in the paper. We do concur that realistic settings involve markets that are "large but finite;" while many of our results carry over to such settings as well (we will remark upon this appropriately in the online appendix), pursuing a full systematic investigation is beyond the scope of this paper and we leave it to future work.
>
>
>
> (II) LOWER BOUNDS. The two lower bounds address different aspects of algorithm design. While Theorem 1 provides general information-theoretic bounds, Theorem 2 provides a uniform lower bound on achievable performance under the most natural policy class $\Pi_m$ for this setting derived using novel ideas based entirely in convex analysis as opposed to traditional exponential tilting techniques. The presence of $\underline{\Delta}_j$ in the numerator here as opposed to the denominator is not an analysis artifact but is, in fact, on account of a fundamentally different complexity of our problem setting vis-\`a-vis classical multi-armed bandits. We have attempted to address this in lines 251-272 but we will revise the exposition to elucidate these differences better.
>
> (III) A REMARK ON ALGORITHMS. The UCB1 algorithm in Figure 2 is run on a two-armed bandit instance while the CAB-2(UCB) algorithm is run on an instance of a countable-armed bandit with two arm-types. The two algorithms are not directly comparable. The plots only intend to highlight that achievable regret scales differently in the two settings. Secondly, while Sampling-UCB does perform better than CAB-K, it also requires the distribution of types $\boldsymbol{\alpha}$ to be specified ex ante. CAB-K, on the other hand, is agnostic to the same. Lastly, as you observed, CAB-K(UCB) is a UCB-based version of CAB-K that performs very well in practice. However, we have an upper bound of $\mathcal{O}\left( \log n \right)$ on its performance only when $K=2$; for $K>2$, the algorithm does achieve sublinear regret but characterizing the best achievable rate remains an open problem.
>
> We have taken due note also of other points you raised and will attempt to address them appropriately in the revision. Thank you again for your time and interest in our manuscript!

---

> > ### Comment · Reviewer_ZuWy · 2022-08-08
> > **Thanks for the response**
> >
> > Thanks for the clarification on the algorithms and experiments. It is very helpful. Regarding to the two versions of lower bound, if there is any change in the main paper, it would be helpful to highlight them in the revised paper.
> >
> > On the other hand, I am not fully convinced by the motivation of the considered problem. I am aware that matching under uncertainty is a recent topic, and most of existing work after [22] focus on stable matching as there are limited agents/arms and there exists competition. This setting is well motivated from real two-sided markets, like job-company, school-student, doctor-hospital. Different from it, this paper is claimed to study large centralized markets where the platform's operational objective is to learn a "maximal" matching (as opposed to stable), and understand how "choice overload" affects achievable performance. As I mentioned in the first review, in real matching problems, we typically have finite markets. It would be important to add a convincing real example to motivate the considered setting. In addition, given such real example, it is also important to discuss why existing dynamic matching methods with finite markets could not handle it.

---

> > > ### Author Response · Authors · 2022-08-08
> > > **Motivation underlying the model**
> > >
> > > Thank you for this question. It is typical in the operations research literature to study choice overload in large markets by essentially considering an auxiliary market setting endowed with an unlimited supply of agents governed by a distribution over "agent-types" (aka the supply distribution in our paper). Essentially the kind of applications we have in mind involve online resource matching/routing/allocation in settings such as freelance markets. Although typical examples would include platforms like TaskRabbit and UpWork, even recommendation systems fit naturally into our paradigm. Please see reference [17] (Matching while Learning (2021) by Johari et al.) for a comprehensive survey of settings with a similar characteristic. Alternatively, one may view our problem as a contextual bandit problem where contexts (jobs) arrive at discrete times in batches of stochastic size and composition, and the learner selects an action (worker) for each arriving context from a countable set. While the traditional linear contextual bandit problem embeds actions in $\mathbb{R}^d$, we do the same in $\mathbb{N}$ (since workers are countable).
> > >
> > > As far as the model itself is concerned, segmenting large populations into finitely many sub-populations based on their "similarities" is a fairly common approach/assumption in analysis. Consider, for example, the case of an online task matching platform like TaskRabbit. The platform typically has sufficient information (courtesy pilot experiments) to know the distribution of worker-skills. For instance, this could be a factoid of the form that there roughly exist only $K$-many possible skill levels in the worker-population for a job descriptor falling in the category of "household electrical repairs." Naturally, the platform's goal in such settings is to maximize the match quality for each arriving job since workers are usually overabundant. This is exactly the maximal matching objective that we consider as opposed to the stable matching one conventionally studied. Also, the stable matching objective is ill-suited to such settings.
> > >
> > > As far as the large market assumption (with unlimited supply) goes, we concur that realistic settings involve markets that are "large but finite." However, we view our model only as a necessary intermediate step towards developing a comprehensive theory for large but finite markets. It is noteworthy that none of the extant works on large or finite markets provides any actionable insight (or algorithm) for our setting because of the restrictive nature of their assumptions; this typically involves endowing the learner with ex ante knowledge of all the key problem primitives. We do not posit any such assumptions in this paper and consequently, doing a full blown analysis is outside the ambit of current work. A large market assumption is therefore necessary for tractability. Of course, translating our results to finite market settings is a very important open problem that we feel is best left to future research at the moment.
> > >
> > > As a quick final remark, although our work has the large market regime as the underlying setting of interest, our learning algorithm CAB-K can readily be adapted to settings that only have finitely many workers in a way that preserves its logarithmic performance guarantees. We will definitely remark upon this in the revision and add relevant pointers in the main text to appropriate sections of the online appendix.
> > >
> > > To conclude, we have duly noted all your flagged concerns and will address them in the revision to the extent permitted by page constraints. Thank you again for the careful reading and the suggestions!

---

### Official Review · Reviewer_Hcpr · 2022-07-10

**Rating:** 6
**Confidence:** 4
**Soundness:** 3 good
**Presentation:** 2 fair
**Contribution:** 3 good

**Summary:**

This paper considers a dynamic matching problem between a stream of jobs and a distribution of workers. Specifically, at each time step, some jobs arrive in a stream where each job has type. The total number of types is finite. The workers are generated according to some distribution $\mathcal{D}$. Under this setting, this paper presents an algorithm that achieves $O(\log(n))$ regret where $n$ is the number of rounds.


**Questions:**


1. State the setting very clearly in Problem Formulation Section. For example: (1) when I first read the paper, I do not know whether the workers’ distribution is known or unknown. Why not state in the 2nd paragraph of Section 2? (2) in line 40 the authors assume $K_j$ is known. This should also be collected in Section 2 when $K_j$ is defined.

2. The authors mentioned in line 126-127 that compared to a line of work in dynamic matching, the main difference is that this paper studies a case where there can be an infinite number of workers. However, from my point of view this is only a minor difference. If my understanding is correct, the major difference is that these papers consider a stability/equilibrium objective, while this paper considers a pure optimization objective. And the reason behind is that those papers study two-sided matching where both sides have their own utility functions, while this paper considers a centralized matching to maximize total revenue, with no individual preference. I think the authors should emphasize this if they agree with the above.
Besides, I think [1] also considered linear function approximation where the utility function takes linear form. I am not sure whether their framework can also incorporate large size of agents. Also check and compare to [2].

[1] Learning Equilibria in Matching Markets from Bandit Feedback, M Jagadeesan, A Wei, Y Wang, M Jordan, J Steinhardt, 2021

[2] Learn to Match with No Regret: Reinforcement Learning in Markov Matching Markets, Y Min, T Wang, R Xu, Z Wang, M Jordan, Z Yang, 2022

3. Besides above, the authors also did not compare to a vast body of work about matching/combinatorial bandits. See for example:[3, 4, 5, 6].

[3] Cesa-Bianchi, N. and Lugosi, G. Combinatorial bandits. Journal of Computer and System Sciences, 78(5):1404– 1422, 2012

[4] Chen, S., Lin, T., King, I., Lyu, M. R., and Chen, W. Combinatorial pure exploration of multi-armed bandits. In Ghahramani, Z., Welling, M., Cortes, C., Lawrence, N. D., and Weinberger, K. Q. (eds.), Advances in Neural Information Processing Systems 27, pp. 379–387. 2014.

[5] Katariya, S., Kveton, B., Szepesvari, C., Vernade, C., and Wen, Z. Stochastic Rank-1 Bandits. In Proceedings of the 20th International Conference on Artificial Intelligence and Statistics, volume 54 of Proceedings of Machine Learning Research, pp. 392–401, Fort Lauderdale, FL, USA, 20–22 Apr 2017.

[6] Rate-Optimal Contextual Online Matching Bandit, Y Li, C Wang, G Cheng, WW Sun


**Limitations:**

No special concerns.

**Strengths And Weaknesses:**

Originality: The paper is quite original. The problem setting is quite novel to my opinion (I also think it has some real-world application).

Quality: The mathematics looks solid to me.

Clarity: This paper’s model is actually quite complicated. Given the current presentation, I think some effort is required to make the paper easier to understand. Specifically, the setting is too complicated and some details can be found here and there in the paper.

Significance: The significance level is fine. But I suggest the paper to make more technical comparison with other related papers in the dynamic matching field.

---

> ### Author Response · Authors · 2022-08-02
> **Response to reviewer Hcpr**
>
> Thank you for your questions and the references. To your Question 2, indeed your interpretation is correct and we would like to point you to our response to Reviewer ZuWy (please see the bullet on Motivation); we will elucidate this better in the revision. Thank you also for the suggested references. The problems studied in cited papers are fundamentally distinct from our setting and in addition, performance bounds therein scale linearly with the size of the market (see, e.g., [1,2]). We will, however, include them in the literature survey to establish more comprehensively distinctions of our model from extant works. We have also duly noted your concerns regarding the exposition and problem formulation section and will address them appropriately in the revision.

---

### Official Review · Reviewer_kLVS · 2022-07-12

**Rating:** 6
**Confidence:** 4
**Soundness:** 4 excellent
**Presentation:** 4 excellent
**Contribution:** 3 good

**Summary:**

This paper studies the problem of learning while matching in a large market, where a centralized platform (e.g., an online freelancing marketplace) matches arriving jobs to a pool of workers with unknown abilities that must be learned. The authors focus on the challenge of “choice overload,” where the number of workers is much larger than the number of jobs, and thus not all workers can be “explored.” They thus model the dynamic matching problem as an infinite MAB problem, where jobs are arm pulls and an infinite reservoir of workers are the arms. They give an optimal instance-dependent algorithm for learning in this setting, achieving logarithmic regret in the number of jobs $T$.

**Questions:**

- Is there a way to get comparable results for an analog of the problem that does not assume discrete clusters?
- I didn’t quite understand how the technical setup under “Supply of workers.” was used later in the paper. In particular, is there a reason the results would not apply if the workers simply have their types drawn from a joint distribution over per-job means $(\mu_1,\ldots,\mu_J)$?
- For the CAB-K algorithm, how many re-initializations are needed in expectation?

**Limitations:**

The authors do discuss the limitations of their theoretical results.

**Strengths And Weaknesses:**

- Strengths:
    - By studying choice overload in an imbalanced market, authors focus on a distinct (and relevant) learning challenge that is not considered in typical models of learning in matching markets.
    - The authors develop a nice technical result about the infinite MAB problems where the learner is oblivious to the reservoir distribution. They show that the dependence on the reservoir distribution can be reduced to a second-order $\log\log T$ term.
    - The exposition of the results is clear and easy to follow.
- Weaknesses:
    - While the initial motivation is grounded in learning in a market setting, it does not appear to be discussed later in the paper. It would be nice to see a discussion of the economic/managerial implications of the technical results.
    - The assumption of discrete clusters seems essential to how the algorithm works. That is, the algorithm would not appear to apply to even a discretization of a continuous distribution (and in practice one would expect distributions of ability to be continuous), since it relies on being able to cleanly separate the clusters in finite samples.

---

> ### Author Response · Authors · 2022-08-02
> **Response to reviewer kLVS**
>
> Thank you for your assessment and the questions! We will attempt to answer your questions below.
>
> (I) Existence of discrete clusters is fundamental to logarithmic performance guarantees. While one can "over-provision" and assume a continuous distribution of types, algorithms built on such premises will have a polynomial regret in $n$, e.g., $\mathcal{O}\left( \sqrt{n} \right)$ and will not be able to adapt to simpler settings where the population might only contain finitely many clusters. Consequently, they will incur polynomial regret even when logarithmic regret is information-theoretically achievable. In fact, there exist works suggesting that adapting to the smoothness of type distributions in a rate-optimal manner is in general impossible; see, e.g., Locatelli et al., Adaptivity to Smoothness in X -armed bandits. We will remark upon this in the revision; thank you for pointing this out!
>
> (II) Your interpretation is absolutely correct! Indeed, the type vector of a worker can be assumed to be drawn from a $\left\lvert \mathcal{J} \right\rvert$-dimensional joint distribution. We will revise the exposition to elucidate this in simpler terms. Thank you for pointing this out!
>
> (III) Consider a simpler setting with only one job-type in the universe. Then, the number of re-initializations needed equals the number of i.i.d. size $K$ (number of worker-types) draws from the population before one obtains a draw containing all distinct types; this is given by $\frac{1}{K!\prod_{i=1}^{K}\alpha_i}$ in expectation. While this constant can potentially be improved under "smarter" algorithms, such extensions would be beyond the scope of current paper and we leave them to future work. Furthermore, the $\frac{1}{K!\prod_{i=1}^{K}\alpha_i}$ term appears only as a constant in the upper bound, i.e., regret does not scale with the type distribution in a first-order sense. This, in a nutshell, is a key finding of our work.
>
> We have duly noted your other concerns as well and will attempt to address them appropriately in the revision.

---

> > ### Comment · Reviewer_kLVS · 2022-08-09
> > **Thanks for the response**
> >
> > (I): Thanks for the clarification here; that is interesting to know!
> >
> > (III): This constant seems large (and perhaps inhibits the practicality of this approach); it would indeed be interesting to know if a similar result could be obtained without such a dependence on K.

---

> > > ### Author Response · Authors · 2022-08-09
> > > **Dependence on K**
> > >
> > > Indeed, the constant term of $\frac{1}{K!\prod_{i=1}^{K}\alpha_i}$ can potentially be improved by modifying CAB-K (Algorithm 2) in the following way: Instead of discarding an entire batch of $K$ arms, one can alternatively only discard those arms that seem to have "repetitive" types. For example, in the case of $K=2$, this would be tantamount to discarding only one of the two arms in the consideration set (in the event that they seem identical-typed) and replacing it with a newly queried arm, as opposed to discarding both the arms and querying two new arms. However, such a modification greatly complicates analysis when $K>2$ owing to the combinatorial number of possibilities. Furthermore, it remains unclear if the proposed modification leads to the best possible dependence w.r.t. $K$. Thus, while it certainly is very important from a practical standpoint, providing a systematic treatment of this aspect remains outside the ambit of current work which primarily is focused on rate-optimality w.r.t. $n$ and delegating the dependence on $\left( \alpha_1,...,\alpha_K \right)$ to sub-logarithmic terms. We would also like to remark that our work is mainly motivated by settings such as TaskRabbit and UpWork, where one should not expect $K$ to be large (and consequently the constant term also will not be very large). For example, in such applications, one could imagine that several groups of agents that are predicted to be bad at a given task can be clumped into one group (as their rewards are all 0, or close to each other). We do realize that additional discussion on this point would be helpful and will therefore include appropriate pointers. Thank you for the excellent question!

---

### Official Review · Reviewer_TzXV · 2022-07-14

**Rating:** 7
**Confidence:** 3
**Soundness:** 3 good
**Presentation:** 4 excellent
**Contribution:** 3 good

**Summary:**

This work focuses on a sequential matching problem in a two-sided platform composed by "jobs" and "workers". It is assumed that the skills of the workers are uncertain, and the supply of workers is unlimited; the latter assumption is to capture the "choice overload" phenomenon. The platform does not have access to the distribution over the attributes of the workers (skills) and to the distribution and mean of the payoffs for each match. The platform faces the following process (without knowledge of the time horizon): jobs arrive sequentially over time (possibly in batches of random size), upon arrival the platform must decide whether to match the job to a worker or not (in case of deciding to match, this can happen with a previous worker now unmatched or a "new" worker). After the match occurs, a random reward is obtained (platform does not have access to the distribution nor the mean). The goal of the platform is to design a policy that maximizes the expected total payoff. The authors consider the standard regret analysis, so equivalently, the goal is to minimize the expected cumulative regret with respect to the clairvoyant policy.

Results. Theorem 1 presents lower bounds for the specific case of one job type and 2 skill levels over the class of "admissible policies". Theorem 2 presents a lower bound (dependent on skills' levels distribution) for the specific case of one job type and 2 skill levels over the class of memoryless policies. Theorem 3 shows the upper bound for the regret of the main algorithm for one job-type (CAB-K) which can composed with another algorithm (MATCH) to obtain a general upper bound in Theorem 4. Other theoretical results and a computational study are given in the Appendix.


**Questions:**

- How does the first lower bound in Theorem 1 relate to the lower bound in (Kalvit and Zeevi, 2020)?

- How Theorem 2 compares to Theorem 1 (i)? is there any relationship between them?

- I have a couple of questions in terms of modeling of matching platforms (which is the main motivation of the paper). Usually, the platform has access to the rewards of the matches, is there any intuition if the regret guarantees can be improved if the platform knows the reward distribution? The availability of unlimited supply seems an unrealistic assumption for a matching market, but the authors claim that the goal is to capture the choice phenomenon: how is the phenomenon reflected in the results that are obtained? Also, the platform may have access to partial information of the skills of workers (e.g. because of ratings of previous assignments), how the regret guarantees improved if we access to estimates of the skills' distributions?

- Unless I am missing something, the clairvoyant benchmark assumes that we can always assign an optimal worker to an arriving job (in Expression (1)). This seems to be a strong assumption given that the proportion of optimal "workers" might be much smaller than the rest of workers. Is there any intuition on the improvement in terms of regret if we assume that the benchmark cannot always match to the optimal worker (for example, by imposing a budget on optimal workers)?

Minor comments:
- Typos in Appendices references: e.g. Page 5 line 215 it should be C not E; Page 6 line 250 it should be D not F; Page 8 line 345, R should be E; Page 9 line 366, S should be F.
- $K$ is assumed to be constant or much smaller than $n$ (horizon)?


**Limitations:**

The authors have adequately addressed the limitations of their work. This work is mainly theoretical, so there is no potential negative societal impact.

**Strengths And Weaknesses:**

Overall, the paper is well-written and easy to follow; I enjoyed reading it. I appreciate the discussions that are given after every result or as an introduction (e.g. on Page 2 "on the complexity of the problem") because it helps the reader to understand the contributions and limitations of the results. To the best of my knowledge on this area, this work considers a more general version of the CAB problem introduced in (Kalvit and Zeevi, 2020) and provide significant technical contributions. At the same time, this work leaves interesting methodological and modeling questions for future work.

---

> ### Author Response · Authors · 2022-08-02
> **Response to reviewer TzXV**
>
> Thank you for your careful reading and the positive feedback! We will attempt to answer your questions below.
>
> (I) Theorem 1 essentially generalizes the lower bound proof of (Kalvit and Zeevi, 2020) to also facilitate deriving minimax lower bounds. In fact, since the original submission, we have been able to strengthen this result further and remove the $\epsilon$-dependence from Theorem 1(ii). Furthermore, the definition of admissible policies is expanded to also include policies whose regret is monotone w.r.t. $\epsilon$, thereby generalizing this policy class substantially. These changes will be incorporated into the revision.
>
> (II) Re the lower bounds, we would like to point your attention to our response to reviewer ZuWy (please see the bullet on Lower bounds).
>
> (III) Consider a simpler setting with only one job-type in the universe. Then, knowing the reward distributions associated with workers is tantamount to knowing the reward distributions in a multi-armed bandit problem. We know that constant regret is achievable in the latter setting. The complexity of the problem lies in the fact that the platform can only observe noisy rewards as opposed to having distribution-level knowledge. Regarding the choice overload aspect, we would like to point your attention to our response to reviewer ZuWy (please see the bullet on Motivation). The fascinating discovery in our work is that choice overload reflects in the upper bounds only in constant-order terms. That is to say, despite a large population of agents, achievable performance in the problem will effectively scale only with the "rank" of the population (number of types) and not with its "size" (distribution of types and the size of the agent pool).
>     Indeed, the platform may have partial information about worker skill proficiencies gathered through historical data; this is exactly our setting. The point to note is that as long as this information is noisy (collection of stochastic reward realizations), logarithmic regret will be unavoidable in the instance-dependent sense.
>
> (IV) Interesting question and this actually pertains to some directions we are currently pursuing where the market is ``large but finite.'' Indeed, in such cases, an optimal worker cannot always be recommended even by the clairvoyant owing to budget constraints. It will be interesting to see how achievable performance scales in settings with budget constraints. The interesting finding in our work is that despite a strict clairvoyant benchmark, achievable regret scales with the distribution of worker-types only in a second-order sense. That is to say, even if the proportion of optimal workers is very small relative to the rest, it does not reflect in the leading-order coefficient of regret.
>
> (V) Yes, $K$ is assumed to be constant and ex ante known to the platform.
>
> We have also duly noted all typographical errors that you pointed out. Thank you again for the careful reading!

---

### Meta-Review · Area_Chair_ZvFf · 2022-08-21

**Recommendation:** Accept
**Confidence:** Certain

**Metareview:**

We thank the authors for their submission.

The paper studies a matching problem between jobs and workers, where jobs arrive sequentially and workers have unknown skills that need to be learned. The number of workers is much larger than the number of jobs ("choice overload"), meaning not all workers can be explored -- a setting seldom considered in bandit literature. The work further presents novel lower and algorithmic upper bounds on the regret.
The paper is clearly written.

**Award:**

No

---

### Decision · Program_Chairs · 2022-09-14

Accept